# Interdependence of Pasha and Drosha for localization and function of the Microprocessor in *C. elegans*

Thiago L. Knittel [1,7], Brooke E. Montgomery[1,7], Kailee J. Reed[1,2], Madeleine C. Chong [1], Ida J. Isolehto[3,4], Erin R. Cafferty [1], Margaret J. Smith[1], Reese A. Sprister[1], Colin N. Magelky[1], Hataichanok Scherman[5], René F. Ketting [3,6] & Taiowa A. Montgomery [1,2] ✉

Primary microRNA (pri-miRNA) transcripts are processed by the Microprocessor, containing the ribonuclease Drosha and its RNA-binding partner DGCR8/Pasha. In a forward genetic screen utilizing a fluorescence-based sensor that monitors pri-miRNA processing in live *Caenorhabditis elegans*, we identify a mutation in the conserved G179 residue adjacent to the namesake W180 of Pasha's WW domain that disrupts pri-miRNA processing. We show that both the G179 and W180 residues are required for Pasha dimerization and Microprocessor assembly. The WW domain also facilitates nuclear localization of Pasha, likely through its role in Microprocessor assembly, which in turn promotes nuclear enrichment of Drosha. Furthermore, depletion of Pasha mislocalizes Drosha to the cytoplasm, and vice versa, while deletion of Pasha's N-terminus causes both proteins to accumulate in nucleoli. Our results reveal a mutual dependency between Pasha and Drosha for their localization in *C. elegans* and highlight the role of Pasha's WW domain in maintaining Microprocessor integrity.

miRNAs are implicated in nearly every biological process and their dysregulation commonly leads to developmental abnormalities and disease[1]. miRNA biogenesis involves two sequential RNA cleavage steps[2]. The first cleavage event occurs within the nucleus where the Microprocessor complex excises the miRNA hairpin from the pri-miRNA transcript to form the pre-miRNA[3–8]. Following transport to the cytoplasm, Dicer removes the terminal loop from the pre-miRNA hairpin to free the miRNA duplex, marking the second cleavage event[9–12]. The miRNA duplex then binds an Argonaute protein, which eliminates one strand and retains the other to serve as a guide for sequence-specific mRNA silencing[9,13].

The Microprocessor, comprising two core proteins, the RNA-binding protein DGCR8/Pasha and the ribonuclease Drosha, identifies pri-miRNA transcripts and initiates miRNA biogenesis[3–5]. Drosha binds at the base of the hairpin, cleaving it to release it from the primary transcript[3,14]. DGCR8/Pasha forms a dimer at the top of the miRNA hairpin, playing a critical, albeit indirect, role in cleavage of the hairpin[6,14]. The precise function of DGCR8/Pasha remains somewhat uncertain. A short peptide of human DGCR8, containing its DROSHA interaction domain, is sufficient to enable the cleavage of specific miRNA hairpins when complexed with DROSHA. However, longer fragments of DGCR8 containing the double-stranded RNA (dsRNA)

[1]Department of Biology, Colorado State University, Fort Collins, CO, USA. [2]Cell and Molecular Biology Program, Colorado State University, Fort Collins, CO, USA. [3]Biology of Non-coding RNA group, Institute of Molecular Biology, Mainz, Germany. [4]International PhD Program on Gene Regulation, Epigenetics and Genome Stability, Mainz, Germany. [5]Department of Biochemistry and Molecular Biology, Colorado State University, Fort Collins, CO, USA. [6]Institute of Developmental Biology and Neurobiology, Johannes Gutenberg University, Mainz, Germany. [7]These authors contributed equally: Thiago L. Knittel, Brooke E. Montgomery. ✉e-mail: tai.montgomery@colostate.edu

binding domains lead to more effective miRNA processing[14]. Consequently, it is probable that DGCR8/Pasha functions both in dsRNA binding, thereby stabilizing the Microprocessor-pri-miRNA complex, and in ensuring proper folding and orientation of Drosha. Dimerization of DGCR8 is also important for accurate processing of pri-miRNA transcripts, underscoring its role in correctly positioning Drosha on the miRNA hairpin[14]. In *C. elegans*, but seemingly not in humans, PASH-1, the *C. elegans* ortholog of DGCR8/Pasha, also serves as a ruler to guide Drosha/DRSH-1 cleavage at the correct position relative to the top of the hairpin[15].

In this work, we developed a reporter system for pri-miRNA processing in *C. elegans*, a genetically tractable whole-animal model, to explore the first step in miRNA biogenesis in vivo. This system comprises a fluorescent sensor that faithfully reflects the cleavage of a pri-miRNA transcript in *C. elegans*. Utilizing the sensor in a forward genetic screen, we identified a mutation in the G179 residue of the WW domain of PASH-1, a region embedded within the RHED domain important for heme binding and protein dimerization[16–18]. Intriguingly, animals with this mutation, or with an engineered mutation in the adjacent namesake W180 residue of the WW domain, are viable despite modest but widespread reductions in miRNA levels. PASH-1 dimerization and Microprocessor integrity is impaired in WW domain mutants, which likely underlie the strong reduction in nuclear enrichment of both PASH-1 and DRSH-1 in these mutants. Knockdown of *pash-1* via RNA interference (RNAi) also disrupts DRSH-1's nuclear enrichment, while *drsh-1* knockdown leads to a reduction in PASH-1's nuclear localization. Thus, correct assembly of the Microprocessor is important for nuclear import or retention of both PASH-1 and DRSH-1. Furthermore, our findings reveal that while a reduction in the nuclear localization of the Microprocessor is tolerable, nuclear exclusion of the Microprocessor results in embryonic or early larval arrest, underscoring a critical role for nuclear activity of the Microprocessor in development.

## Results

### A sensor for primary miRNA processing in *C. elegans*

We developed a sensor for pri-miRNA recognition and processing in live *C. elegans* that can be monitored on growth media without the need for mounting or immobilization. The sensor contains the hairpin and regulatory sequences of miR-58/bantam[19] fused to mCherry coding sequence, expressed under the ubiquitin, *ubl-1*, promoter and integrated into the genome as a single-copy transgene (Fig. 1a)[20]. When the miR-58 hairpin is cleaved by the Microprocessor, the mCherry mRNA is detached from the *mir-58* 3' regulatory elements, which presumably leads to its degradation since Microprocessor cleavage of exonic hairpins leads to mRNA destabilization[21]. Thus, we predicted that the sensor would produce elevated levels of mCherry if pri-miRNA recognition or processing was impaired. Indeed, mCherry was weakly expressed on vector control RNAi treatment but was strongly expressed on *pash-1* RNAi (Fig. 1b). In contrast, a control strain with *ubl-1* 3' UTR sequence in place of pri-miR-58 sequence displayed similar mCherry expression on control or *pash-1* RNAi (Fig. 1b).

*pash-1* and *drsh-1* are both essential but first-generation mutants are viable[5]. Thus, we introduced the sensor into *drsh-1* mutants, homozygosed the sensor, and then imaged animals segregating *drsh-1+/+* and *drsh-1−/−* in the first generation of homozygosity in which *drsh-1*-deficient animals are still healthy. The sensor was strongly desilenced in *drsh-1−/−* mutants relative to *drsh-1+/+* animals (Fig. 1b). Therefore, the sensor is responsive to loss of either *pash-1* or *drsh-1*, the core components of the Microprocessor. The sensor restored miR-58 levels in *mir-58−/−* mutants to ~11% of wild-type levels but had a negligible impact on miR-58 levels in *mir-58+/+* animals, indicating that the miR-58 hairpin is further processed into mature miR-58 following Microprocessor cleavage, albeit at much lower levels than the endogenous miR-58 gene (Fig. 1c). The sensor has the advantage over other sensors[22] in that it reports on pri-miRNA processing in whole, live animals and can be scored on a standard stereo microscope.

In a proof-of-principle forward genetic screen for cis-acting mutations that disrupt sensor processing, we identified a mutation within the miRNA duplex region of the hairpin (Supplementary Fig. 1a). This mutation caused a modest increase in mCherry expression, which we confirmed by introducing it into non-mutagenized animals using site-directed mutagenesis (Supplementary Fig. 1b). The mutation converts a G-C base pair to a G-U base pair adjacent to the internal loop, likely expanding the loop and interfering with Microprocessor recognition or processing. That this subtle perturbation disrupts processing of the sensor highlights its sensitivity in accurately reflecting the recognition and cleavage of the miR-58 hairpin by the Microprocessor machinery.

### PASH-1's WW domain aids pri-miRNA processing

We then did a forward genetic screen for trans-acting mutations that desilenced the pri-miR-58 sensor. From a non-exhaustive screen of ~40,000 haploid genomes, we selected 72 candidates representing at least 20 independent lines that desilenced mCherry. Of these, only 17 were fertile over multiple generations. Among the fertile lines, 38a displayed the most robust increase in mCherry expression (Fig. 2a, b). We subjected this line to whole genome sequencing and identified a missense mutation in *pash-1* (Fig. 2c). We backcrossed the line to the original non-mutated strain and confirmed that mCherry desilencing tracked with the *pash-1* allele, indicating that it was likely the causal mutation. We then outcrossed the allele, *pash-1(ram33*[G179R]*)*, to wild-type animals three times to remove the sensor and reduce background mutations.

The *pash-1(ram33*[G179R]*)* mutation changes the glycine (G179) residue within PASH-1's WW domain to an arginine (R) (Fig. 2c). This residue is adjacent to the first tryptophan residue (W180) of the WW domain which is embedded in the broader RHED domain that binds heme in humans. Notably, the second W residue found in most WW domains is absent in Pasha/PASH-1 in *D. melanogaster* and *C. elegans* (Fig. 2d)[23]. The G179 residue mutated in *pash-1(ram33*[G179R]*)* is highly conserved in DGCR8/Pasha and other WW domain-containing proteins (Fig. 2d)[23]. Additionally, the WW domain shows near-identical structure when comparing an X-ray diffraction model of human DGCR8 with an AlphaFold3-predicted structure of *C. elegans* PASH-1 (Fig. 2e)[17].

When grown at 20 °C, pri-miR-58 levels were ~2-fold higher in *pash-1(ram33*[G179R]*)* mutants compared to wild-type, as assessed by quantitative real-time PCR (qRT-PCR) (Fig. 2f). At 25 °C, pri-miR-58 levels increased ~5-fold, indicating partial temperature sensitivity of the allele (Fig. 2f). Direct comparison of pri-miR-58 levels between 20 °C and 25 °C was complicated by potential differences in housekeeping gene expression used for normalization. However, without normalization, fold-changes in pri-miR-58 levels between mutants and wild-type were nearly identical to those obtained after normalization to actin, supporting the validity of direct comparisons (Fig. 2g). From this, we observed an ~2-fold increase in pri-miR-58 levels at 25 °C relative to 20 °C, suggesting either increased transcription or reduced processing efficiency at higher temperatures (Fig. 2g). Despite the increase in pri-miR-58 levels, mature miR-58 levels were largely unchanged at 20 °C and were reduced by only ~25% at 25 °C in mutants relative to wild-type when normalized to the unrelated small RNA, 21UR-1 (Fig. 2h). Similar results were obtained without normalization (Fig. 2i). Unlike pri-miR-58, mature miR-58 levels were similar between 20 °C and 25 °C in wild-type animals, suggesting a non-linear relationship between primary and mature miR-58 levels (Fig. 2i). pri-miR-35, pri-miR-51, pri-miR-80, and pri-miR-238 were also all significantly upregulated in *pash-1(ram33*[G179R]*)* mutants at both 20 °C and 25 °C (Supplementary Fig. 2a, b).

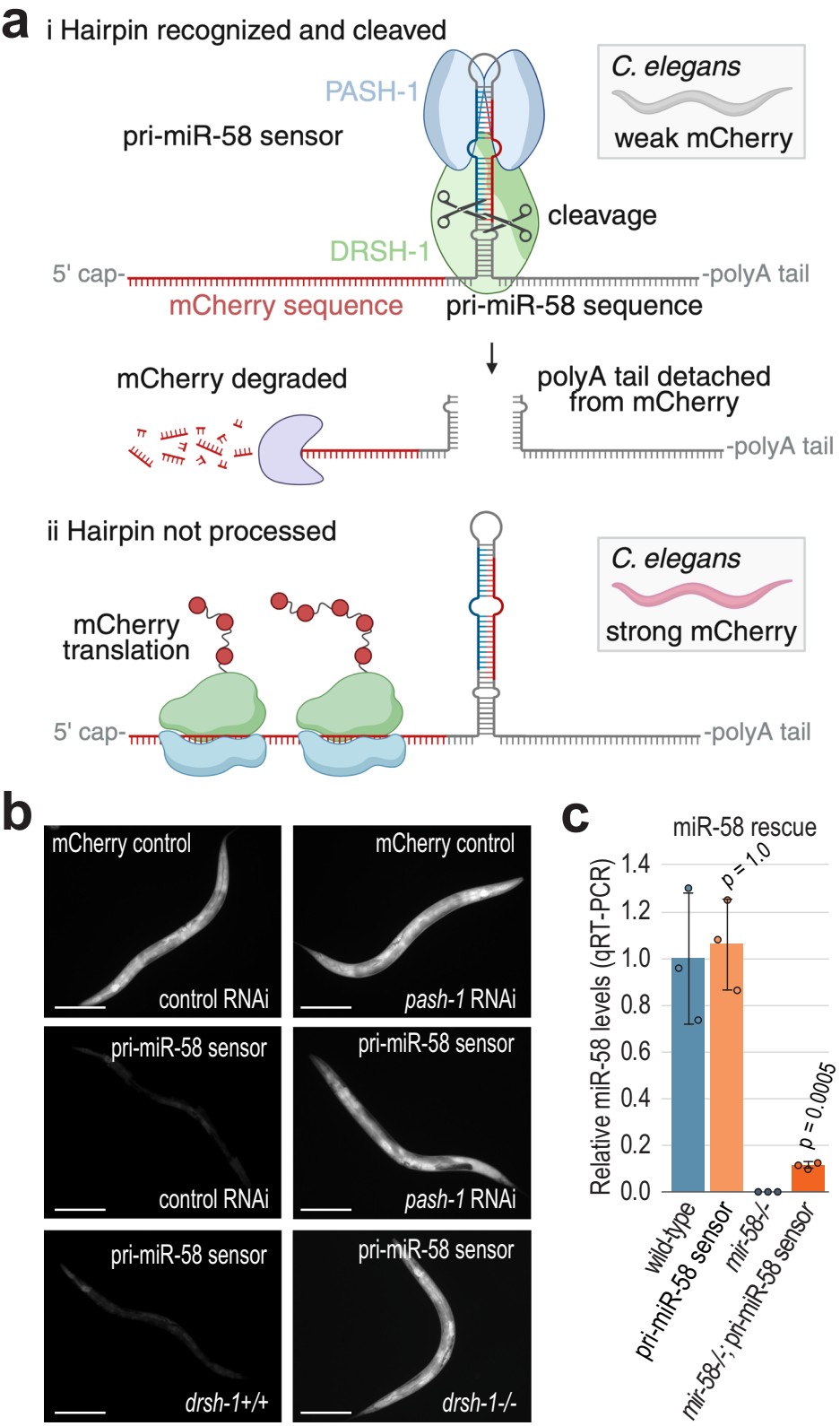

**Fig. 1 | A sensor for pri-miRNA recognition and processing in *C. elegans*. a** pri-miR-58 sensor design. (i) In the presence of a functional pri-miRNA biogenesis pathway, the miR-58 hairpin is cleaved from the mCherry mRNA by the Microprocessor leading to the degradation of mCherry. (ii) If the sensor is not recognized and processed, mCherry is expressed. Created in BioRender. Montgomery, T. (2025) https://BioRender.com/e89g547. **b** mCherry fluorescence in animals containing the pri-miR-58 sensor or a control construct lacking pri-miR-58 sequence (mCherry control). Animals were treated with either control (empty L4440 vector) or *pash-1* RNAi or were segregants for wild-type *drsh-1* or the *drsh-1* deletion allele *drsh-1(ok369)*. Scale bars = 0.1 mm. At least three representative individuals were imaged for each condition. **c** Relative levels of mature miR-58 normalized to let-7 in the various strains indicated as determined by TaqMan qRT-PCR. Error bars are standard deviation (SD) from the mean. *n* = 3 biological replicates. Two-tailed, two-sample Student's t-tests were used to calculate *p*-values for comparisons to wild-type. A Bonferroni correction for three comparisons was applied. Source data are provided as a Source Data file.

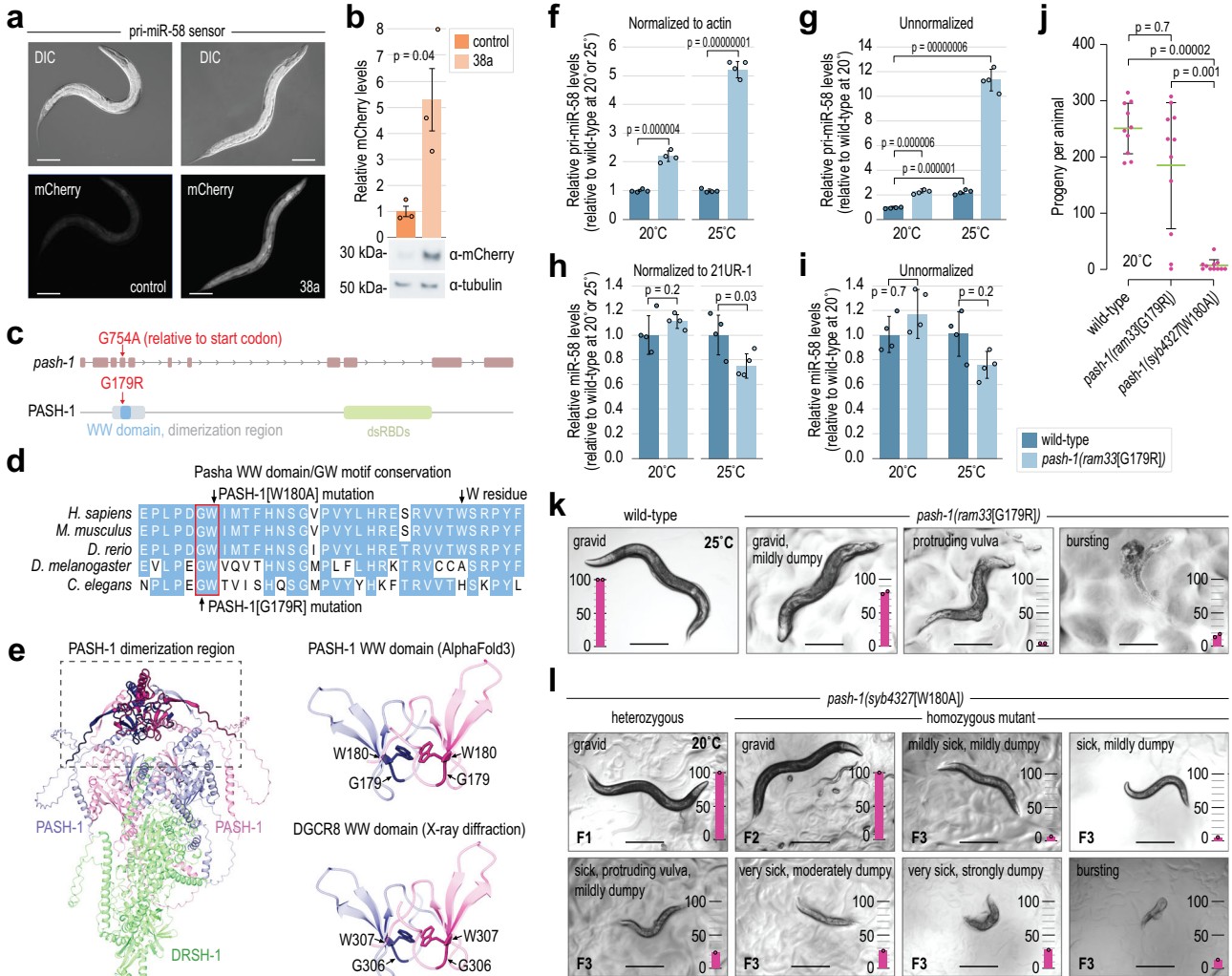

**Fig. 2 | Requirement for PASH-1's WW domain in pri-miRNA recognition or processing. a** Representative images of control and 38a mutants containing the pri-miR-58 sensor. Scale bars = 0.1 mm. At least ten embryos per condition were imaged. **b** Quantification of mCherry levels in non-mutant control and 38a mutants by Western blot. One of 3 representative blot images is shown (see Source Data file). Tubulin was used for normalization. Error bars are SD. $n = 3$ biological replicates. A two-tailed, two-sample Student's t-test was used for statistical analysis. **c** Location of the 38a mutation on *pash-1* DNA and protein. dsRBDs, double-stranded RNA-binding domains. **d** WW domain sequence alignment. **e** AlphaFold3-predicted structure of the *C. elegans* Microprocessor highlighting the WW domain (residues 174–207) and dimerization region (residues 148-266). Human X-ray diffraction-based structure of the DGCR8 WW domain (PDB: 3LE4) is shown for comparison. **f**, **g** Relative endogenous pri-miR-58 levels in wild-type and *pash-1(ram33*[G179R]*)* mutants grown at 20 °C or 25 °C as measured by TaqMan qRT-PCR and normalized to *act-1* (**f**) or unnormalized (**g**). Error bars are SD. $n = 4$ biological replicates. Two-

tailed, two-sample Student's t-tests were used for statistical analysis. A Bonferroni correction for three comparisons was applied in (**g**). **h**, **i** Relative mature miR-58 levels in wild-type and *pash-1(ram33*[G179R]*)* mutants grown at 20 °C or 25 °C as measured by TaqMan qRT-PCR and normalized to 21UR-1 (**h**) or unnormalized (**i**). Error bars are SD. $n = 4$ biological replicates. Two-tailed, two-sample Student's t-tests were used for statistical analysis. A Bonferroni correction for three comparisons was applied in (**i**). **j** Numbers of progeny produced by animals grown at 20 °C. Error bars are SD. $n = 10$ (wild-type) or 11 (*pash-1(ram33*[G179R]*)* and *pash-1(syb4327*[W180A]*)*) individuals. Two-tailed Mann-Whitney U tests were used for statistical analysis. **k**, **l** Images of wild-type and *pash-1(ram33*[G179R]*)* (**k**) or *pash-1(syb4327*[W180A]*)* (**l**) animals grown at 20 °C or 25 °C as indicated. Bar plots show percentages of animals with the indicated phenotypes ($n = 100$ animals per strain). Images approximate phenotypes scored. Two (**k**) or 1 (**l**) independent experiments were done. Scale bar = 0.3 mm. Source data are provided as a Source Data file.

## Mutations in PASH-1's WW domain cause developmental defects

Animals harboring the *pash-1(ram33*[G179R]*)* mutation are viable, and when grown at 20 °C tended to produce nearly as many progeny as wild-type, but with a higher incidence of animals producing very few progeny (Fig. 2j). When grown at 25 °C, *pash-1(ram33*[G179R]*)* mutants consistently produced fewer progeny than wild-type and ~50% were sterile (Supplementary Fig. 2c). Nevertheless, even at 25 °C, most animals appeared remarkably normal aside from a slight squatty (i.e., dumpy) phenotype, although some had protruding vulvas or extrusion of their guts through their vulvas (i.e., bursting), which are phenotypes common to *C. elegans* miRNA mutants (Fig. 2k)[9].

Attempts to recover mutations in the WW domain of Pasha in flies have failed, likely due to lethality[24]. Thus, we were surprised to find that G179R mutants appeared relatively healthy. It is possible that the mutation only partially impairs the WW domain's function. Therefore, to determine if the WW domain is essential for development, we mutated the W180 residue to alanine (A) using CRISPR-Cas9 genome editing (Fig. 2d)[25–28]. The pri-miR-58 sensor was desilenced when introduced into this mutant, demonstrating that the W180 residue is required for efficient pri-miRNA processing (Supplementary Fig. 2d). Unlike G179R mutants, W180A mutants were nearly sterile at 20 °C and completely sterile at 25 °C (Fig. 2j and Supplementary Fig. 2c). W180A mutants also displayed a spectrum of severe developmental defects,

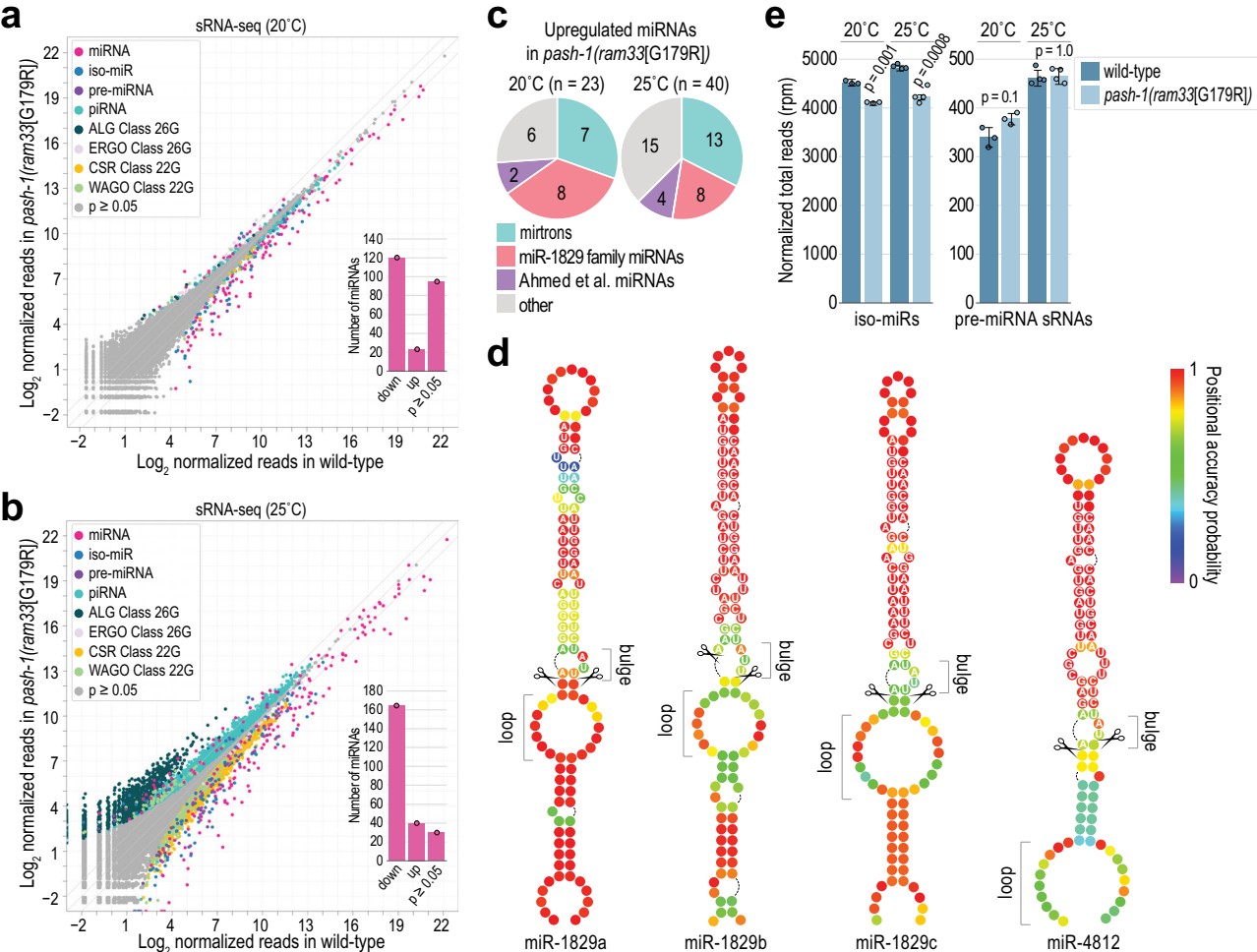

**Fig. 3 | Widespread reduction in canonical miRNA levels in *pash-1(ram33*[G179R]*)* mutants. a, b** Log$_2$ average geometric-mean normalized sRNA-seq read counts in wild-type (x-axis) and *pash-1(ram33*[G179R]*)* (y-axis) animals grown at 20 °C (**a**) or 25 °C (**b**). Small RNA features are represented by data points colored by their classification. Diagonal lines show 0-, 2-, and -2-fold enrichments. The Wald test was used for statistical analysis. The inset bar plots show the numbers of miRNAs represented by >20 geometric mean-normalized reads significantly down or upregulated ($p < 0.05$) or unchanged ($p \geq 0.05$) in *pash-1(ram33*[G179R]*)* relative to wild-type (no fold-change cutoff was applied). $n = 3$ (**a**) or 4 (**b**) biological replicates per strain. **c** Classification of miRNAs upregulated ($p < 0.05$) in *pash-1(ram33*[G179R]*)* relative to wild-type animals grown at either 20 °C or 25 °C, as indicated, based on data in (**a**) and (**b**). **d** Secondary structure predictions of miR-1829 family miRNA hairpins. Nucleotide sequences are shown for the miRNA duplex regions. The cleavage sites are indicated with scissors. **e** Mean reads per million (rpm)-normalized sRNA-seq counts for iso-miRs (offset by ±1–3 nt relative to miRNA 5′ end) and small RNAs derived from pre-miRNAs (pre-miRNA derived reads offset at their 5′ ends by >3 nt relative to mature miRNA 5′ end) in wild-type and *pash-1(ram33*[G179R]*)* mutants grown at 20 °C or 25 °C. Data as in (**a**) and (**b**). Error bars are SD. $n = 3$ (20 °C) or 4 (25 °C) biological replicates per strain. Two-tailed, two-sample Student's t-tests were used to calculate $p$-values for comparisons to wild-type. A Bonferroni correction for two comparisons was applied to each. Source data are provided as a Source Data file.

including dumpy, protruding vulva, and bursting phenotypes (Fig. 2l). These defects were absent in the first generation of homozygosity of the W180A mutation (F2), suggesting that a maternal contribution of wild-type PASH-1 or processed miRNAs supports normal development (Fig. 2l). Thus, while W180 is not critical for viability, it is essential for normal development.

Although the W residue mutated here is a defining feature of WW domains, we investigated whether the G179 and W180 residues, collectively referred to as the GW motif, have additive functions by generating a strain with both mutations (GW-mut). These double mutants produced significantly fewer progeny than either single mutant (Supplementary Fig. 2e; mutations were made in the *pash-1::GFP* background described below). This suggests that G179 and W180 contribute additively to the WW domain's function.

### Widespread loss of miRNAs in PASH-1 WW domain mutants

To globally assess the impact of the *pash-1(ram33*[G179R]*)* mutation on mature miRNA levels, we subjected wild-type and mutant adults grown at 20 °C and 25 °C to small RNA high-throughput sequencing (sRNA-seq). At 20 °C, most canonical miRNAs were significantly downregulated (1.2–16.5-fold) in mutants relative to wild-type (Fig. 3a; Supplementary Data 1). At 25 °C, miRNA levels were further reduced, although developmental defects at this temperature limit interpretation of individual miRNAs (Fig. 3b; Supplementary Data 1). Mirtrons, which bypass Microprocessor processing, were upregulated in mutants at both temperatures (Fig. 3c; Supplementary Data 1)[29,30]. Additionally, the miR-1829 family (4 miRNAs) and miR-42 cluster (2 miRNAs) were upregulated at 20 °C (Fig. 3c; Supplementary Data 1). However, the miR-42 cluster was depleted at 25 °C, while the miR-1829 family was elevated at both temperatures (Supplementary Data 1). miR-1829 family members may be poor Microprocessor substrates due to bulges at their hairpin cleavage sites and large loops in their lower stem (Fig. 3d). If these miRNAs are Microprocessor-independent, their processing mechanism is unclear. Several miRNAs identified by Ahmed et al. were also upregulated and thus may also bypass Microprocessor processing (Fig. 3c; Supplementary Data 1)[31]. Misprocessing

of pri-miRNAs in *pash-1(ram33*[G179R]*)* mutants could free downstream resources, such as Dicer or Argonaute, which may be redirected to mirtrons and other Microprocessor-independent miRNAs. Conversely, miRNAs that are less sensitive to the G179R mutation but still dependent on the Microprocessor could be upregulated due to reduced competition with other pri-miRNAs.

Dimerization of DGCR8 via the RHED region, which contains the WW domain, has been proposed to play a role in miRNA processing precision, based on observations that Microprocessor complexes containing only a single molecule of human DGCR8 process miRNA hairpins with reduced accuracy[14]. Thus, the reduction in mature miRNA levels observed in *pash-1(ram33*[G179R]*)* mutants could result from erroneous cleavage, rather than a loss of pri-miRNA processing. However, we did not observe an increase in the abundance of miRNA isoforms (iso-miRs) shifted by 1-3 nucleotides (nts) at their 5' ends relative to the annotated miRNA locus in *pash-1(ram33*[G179R]*)* mutants (Fig. 3a, b; Supplementary Data 1). Instead, total iso-miR reads were slightly lower in mutants, consistent with the overall decline in canonical miRNA levels (Fig. 3e). Furthermore, we detected only minor differences in sRNA-seq reads originating from pre-miRNA sequences that neither correspond to mature miRNAs nor to the iso-miRs described above, which could occur in more extreme cases of erroneous cleavage (Fig. 3a, b; Supplementary Data 1). There was a slight increase in the total level of these pre-miRNA sRNA-seq reads in *pash-1(ram33*[G179R]*)* mutants grown at 20 °C, but not at 25 °C (Fig. 3e). Additionally, the desilencing of the sensor in the WW domain mutants suggests that these mutations lead to a loss of cleavage rather than altering cleavage precision, as cleavage at any position within the pri-miR-58 sensor's hairpin would presumably result in desilencing (Fig. 2a and Supplementary Fig. 2d). These results suggest that the G179 residue, and by extension the WW domain, is unlikely to play a major role in pri-miRNA processing precision in *C. elegans*. Instead, it appears to have a broader function, although not necessarily directly, in pri-miRNA recognition or cleavage.

Developmental defects and inconsistencies in developmental progression precluded meaningful analysis of global miRNA levels in *pash-1(syb4327*[W180A]*)* mutants. However, by qRT-PCR we found that miR-1 and miR-58 were modestly depleted in *pash-1(syb4327*[W180A]*)* mutants, with the caveat that we did not control for animal morphology (Supplementary Fig. 2f). The modest depletion observed here is consistent with observations in HeLa cells containing a W > A substitution at the second W residue of DGCR8's WW domain and point to a similar requirement for the WW domain in miRNA processing in humans and nematodes[22].

We conclude that while PASH-1's G179 and W180 residues are necessary for optimal miRNA biogenesis, they are not strictly essential. Furthermore, the relatively normal morphology of *pash-1(ram33*[G179R]*)* mutants and the viability of *pash-1(syb4327*[W180A]*)* mutants, despite significantly lower miRNA levels, indicate that *C. elegans* has a high tolerance for reduced miRNA activity. This tolerance might explain why the loss of most individual miRNAs does not result in strong developmental defects[32].

## Cytoplasmic and nuclear localization of PASH-1 and DRSH-1

To investigate the subcellular localization of the *C. elegans* Microprocessor and to determine if it differs in G179R and W180A mutants, we first introduced GFP as a C-terminal fusion with PASH-1 and mCherry as an N-terminal fusion with DRSH-1 through CRISPR-Cas9-mediated editing of their genomic loci[25–28]. Mature miRNA levels were not reduced in these strains, indicating that the modified proteins maintain their normal functions (Supplementary Fig. 3a). Both PASH-1::GFP and mCherry::DRSH-1 proteins localized to the nucleus, as previously observed for their human counterparts (Fig. 4a, b)[33–35]. Consistent with observations in human cells[33,36], we occasionally saw nucleolar enrichment of PASH-1::GFP and mCherry::DRSH-1, particularly in oocytes (Fig. 4a, b).

We then combined the *pash-1::GFP* and *mCherry::drsh-1* alleles into a single strain to examine protein co-localization. PASH-1::GFP and mCherry::DRSH-1 co-localized throughout embryogenesis and promptly reentered the nucleus after cell division, appearing to do so simultaneously (Fig. 4c and Supplementary Fig. 3b). However, they occasionally partitioned differently within nuclear foci (Supplementary Fig. 3c). While both proteins were predominantly localized to the nucleus, they also had relatively strong cytoplasmic signals, although the signal from PASH-1::GFP, but not mCherry::DRSH-1, was somewhat masked by background fluorescence (Fig. 4c, d). Based on planar fluorescence quantification, the average signal of mCherry::DRSH-1 was ~2–4 fold greater in the nucleus compared to the cytoplasm (Fig. 4d, e). These results demonstrate that nuclear enrichment of the Microprocessor is conserved in *C. elegans* but point to the existence of both nuclear and cytoplasmic fractions.

## PASH-1 mislocalizes to the cytoplasm in WW domain mutants

Next, we asked if the G179R and W180A mutations disrupt subcellular localization of PASH-1 by introducing GFP as a C-terminal fusion at the mutant *pash-1* loci. We detected PASH-1::GFP, as well as the G179R and W180A mutant forms, in the nucleus and cytoplasm of germline and embryonic cells (Fig. 4f, g). However, the strong nuclear-to-cytoplasmic enrichment we observed for PASH-1::GFP was lost in both the G179R and W180A mutants, indicating that these residues are important, presumably indirectly, for nuclear entry or retention of PASH-1 (Fig. 4f, g). We did not observe a further reduction in nuclear signal in the strain containing both the G179R and W180A mutations, indicating that there is not an additive effect of the two mutations on PASH-1 localization (Supplementary Fig. 3d).

In *Drosophila*, Pasha's WW domain promotes association with RNA Polymerase II (Pol II), thereby coupling pri-miRNA transcription to miRNA processing[24]. Hence, Pol II could promote nuclear retention of PASH-1 in *C. elegans*, which might explain why its nuclear localization is lost in G179R and W180A mutants. To test this, we first assessed whether deletion of *cdk-9*, a kinase that phosphorylates the C-terminal domain (CTD) of Pol II and which is required for association of Pasha with Pol II in *Drosophila*, desilenced the pri-miR-58 sensor[24,37]. We observed a modest increase in mCherry expression in first generation animals homozygous mutant for *cdk-9*, relative to their heterozygous counterparts, although loss of *cdk-9* leads to larval arrest, which could confound these results (Supplementary Fig. 4a)[38]. While this suggests that *cdk-9* may have a role in pri-miRNA processing, because CDK-9 regulates Pol II activity, desilencing of the sensor could also be caused by reduced expression of *pash-1* or *drsh-1*, which we did not explore further. We then did RNAi against *cdk-9* to assess whether loss of *cdk-9* affects PASH-1::GFP localization. Neither of two different RNAi clones, both of which caused potent embryonic arrest indicative of *cdk-9* knockdown, disrupted PASH-1::GFP localization, suggesting that phosphorylation of Pol II's CTD is not required for nuclear retention of PASH-1 (Supplementary Fig. 4b).

To more directly assess a requirement of Pol II for PASH-1's localization, we did RNAi against the large subunit of Pol II, *ama-1*[39]. RNAi treatment against *ama-1* using dsRNA-expressing bacteria diluted 5x led to embryonic arrest and retention of embryos *in utero* presumably because of the essential role of Pol II in transcription (Supplementary Fig. 4c). A 10x dilution of the RNAi treatment still caused embryonic arrest but arrested eggs were laid on the plate (Supplementary Fig. 4c). Neither treatment caused PASH-1::GFP to mislocalize to the cytoplasm (Supplementary Fig. 4c). Furthermore, we were not able to detect an interaction between PASH-1::GFP and Pol II by protein co-immunoprecipitation (co-IP) (Supplementary Fig. 4d). Therefore, we conclude that PASH-1's nuclear localization is not facilitated by an association with Pol II and thus that the G179R and W180A mutants do not mislocalize due to loss of Pol II association with the Microprocessor.

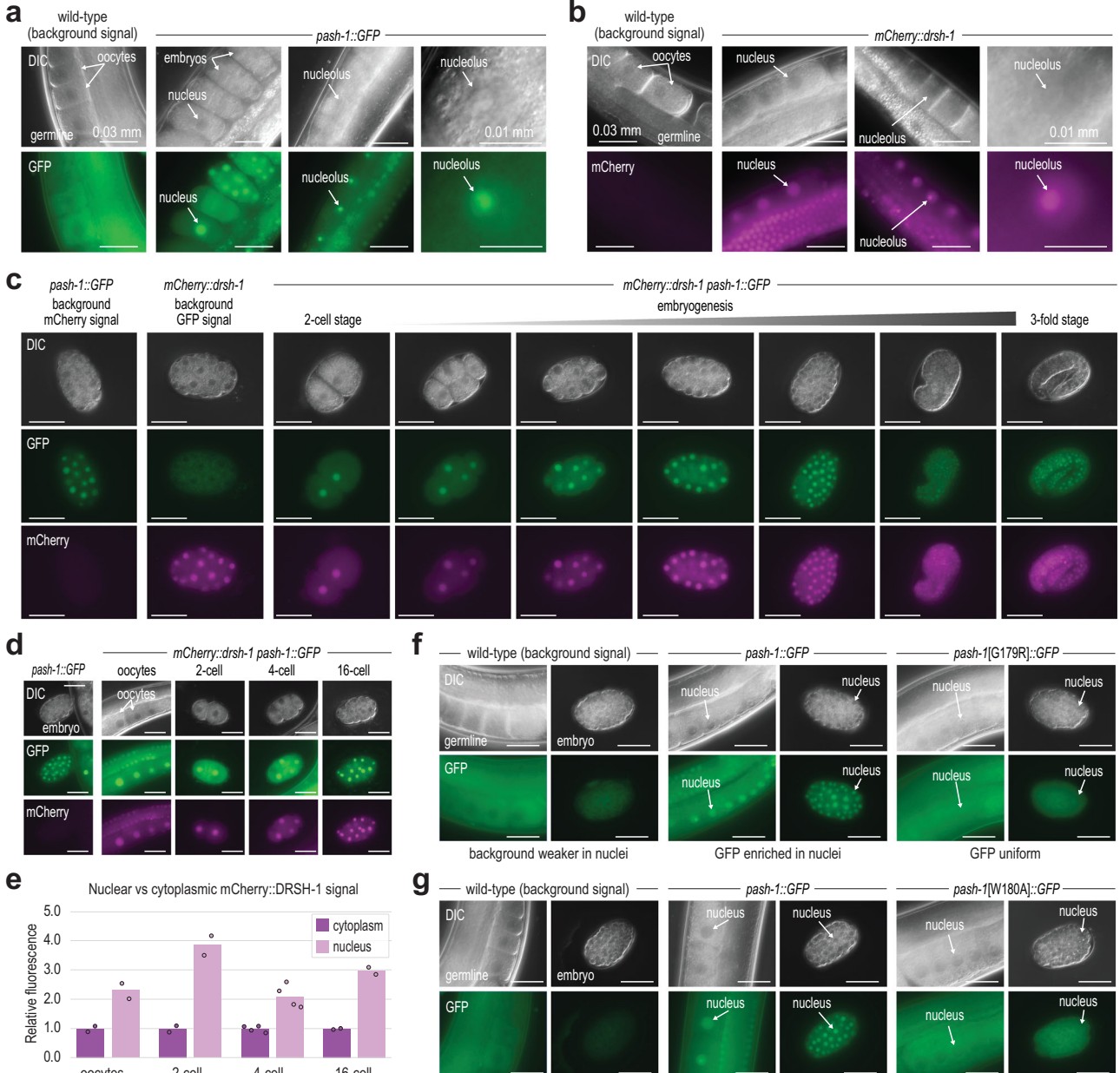

**Fig. 4 | PASH-1 WW domain-mutant proteins mislocalize to the cytoplasm.**
**a** PASH-1::GFP expression in the adult germline and embryos (*in utero*). A wild-type animal is shown to highlight the non-specific background signal observed in germline tissue. The scale bars are 0.03 or 0.01 mm as indicated. At least three representative individuals for each strain were imaged. Imaging was repeated in three independent experiments with at least 10 individuals in total for each strain imaged. **b** mCherry::DRSH-1 expression in the adult germline. A wild-type animal is shown to highlight the lack of non-specific background signal observed in germline tissue. The scale bars are 0.03 or 0.01 mm as indicated. At least three representative individuals for each strain were imaged. **c** PASH-1::GFP and mCherry::DRSH-1 expression in 2-cell through 3-fold stage embryos. Strains lacking either PASH-1::GFP or mCherry::DRSH-1 are shown as controls for background fluorescence. The scale bars are 0.03 mm. At least three representative embryos for each stage were

imaged. **d** PASH-1::GFP and mCherry::DRSH-1 expression in oocytes and 2, 4, and 16-cell embryos. A PASH-1::GFP embryo lacking mCherry::DRSH-1 is shown to highlight the lack of non-specific background signal observed for mCherry::DRSH-1 in embryos. Note both nuclear and cytoplasmic signal. The scale bars are 0.03 mm. At least two representative embryos or germlines were imaged for each strain. Imaging was repeated three times with at least 10 individuals for each strain in total imaged. **e** Relative mCherry fluorescence signal in the nucleus or cytoplasm of the embryos in (**d**). $n = 2$ (oocytes, 2-cell, and 16-cell) or 4 (4-cell) biological replicates. **f**, **g** PASH-1::GFP and PASH-1[G179R]::GFP (**f**) or PASH-1[W180A]::GFP (**g**) expression in the adult germline and embryos. Wild-type germlines and embryos are shown to highlight the non-specific background signal. The scale bars are 0.03 mm. At least fourteen representative embryos or germlines for each strain were imaged. Source data are provided as a Source Data file.

## Interdependence of PASH-1 and DRSH-1 on nuclear localization

To determine if the G179R and W180A mutations in PASH-1 also affect DRSH-1's localization, we introduced *mCherry::drsh-1* into the strains containing *pash-1[G179R]::GFP* or *pash-1*[W180A]*::GFP*. In both mutants, mCherry::DRSH-1 lost its nuclear enrichment (Fig. 5a). By Western blot, we observed ~50% lower levels of GFP in animals containing *pash-1[G179R]::GFP* or *pash-1*[W180A]*::GFP* compared to

animals with unaltered *pash-1::GFP* indicating that the G179R and W180A mutations partially destabilize PASH-1 (Supplementary Fig. 5a). In contrast, mCherry::DRSH-1 levels were only mildly reduced in animals containing the *pash-1* mutations (Supplementary Fig. 5a).

It is possible that the G179R and W180A mutations cause PASH-1 to misfold or aggregate, which in turn could cause DRSH-1 to aggregate, as DROSHA aggregates when overexpressed in human cells in the

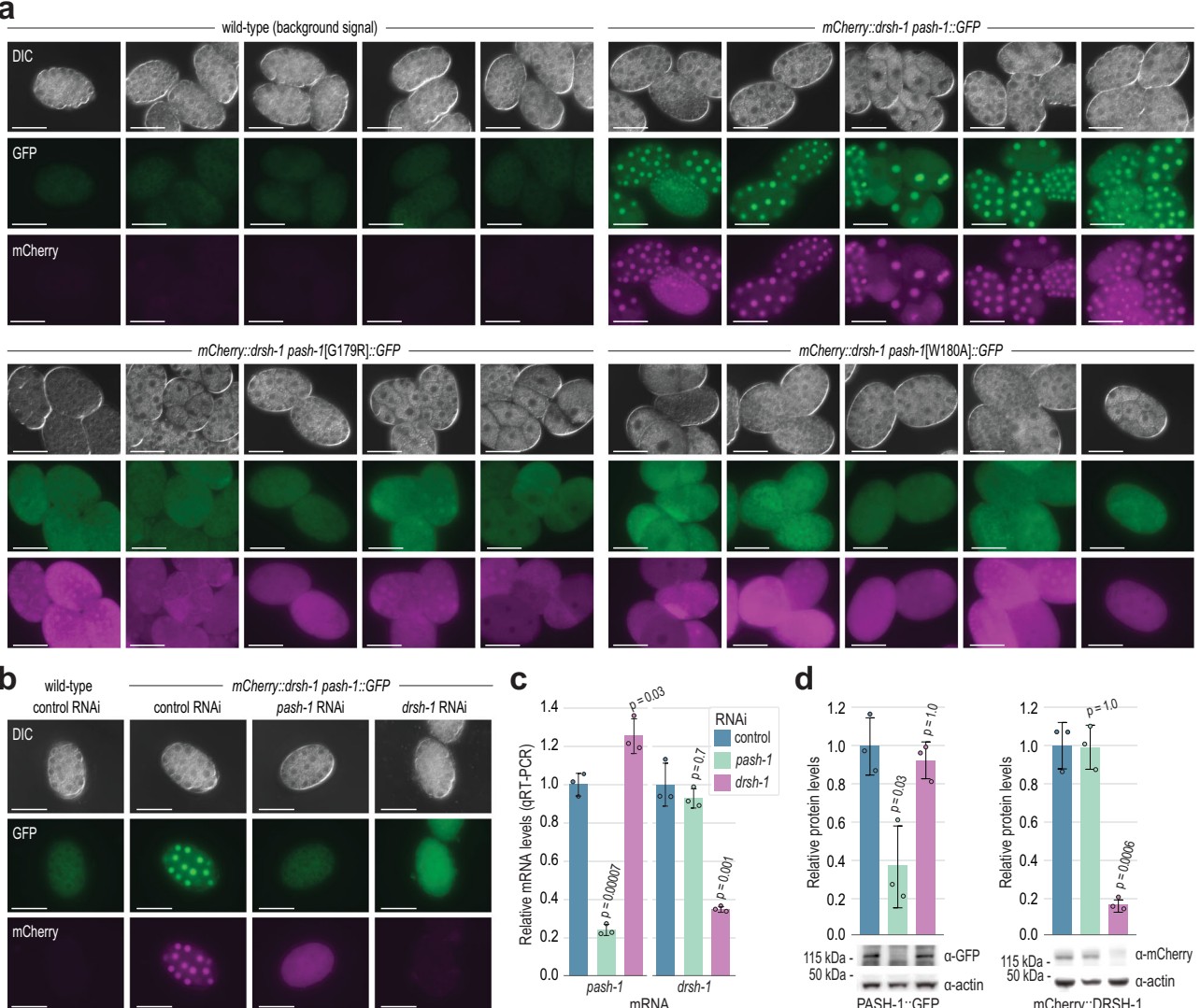

**Fig. 5 | PASH-1 and DRSH-1 promote each other's nuclear localization. a** PASH-1::GFP, PASH-1[G179R]::GFP, PASH-1[W180A]::GFP, and mCherry::DRSH-1 expression in embryos. Wild-type embryos show the non-specific background signal. The scale bars are 0.03 mm. At least twenty embryos were imaged for each strain. **b** PASH-1::GFP and mCherry::DRSH-1 expression in embryos following control (L4440), *pash-1*, and *drsh-1* RNAi. Wild-type embryos show the non-specific background signal. The scale bars are 0.03 mm. In the experiment shown, >100 animals for each condition were monitored, and 1 representative embryo was imaged. The experiment was repeated three times, with a total of at least 23 embryos imaged for each condition. **c** Relative *pash-1* and *drsh-1* mRNA levels following control (L4440), *pash-1*, and *drsh-1* RNAi as determined by qRT-PCR. *rpl-32* mRNA levels were used for normalization. Error bars are SD. *n* = 3 biological replicates. Two-tailed, two-sample Student's t-tests were used to calculate *p*-values for comparisons to control RNAi. A Bonferroni correction for two comparisons was applied. **d** Relative PASH-1::GFP and mCherry::DRSH-1 protein levels after *pash-1* or *drsh-1* RNAi. Actin was used for normalization. Error bars are SD. One of 3 representative blot images is shown (see Source Data file). *n* = 3 biological replicates. Two-tailed, two-sample Student's t-tests were used to calculate *p*-values for comparisons to control RNAi. A Bonferroni correction for two comparisons was applied. Bars are colored as in (**c**). Source data are provided as a Source Data file.

absence of compensatory DGCR8 overexpression[14,40]. We did not observe an increase in PASH-1::GFP or mCherry::DRSH-1 puncta in these mutants, suggesting that the proteins maintain their diffuse cellular expression and do not form large aggregates (Fig. 5a). Given that PASH-1 and DRSH-1 are relatively large proteins (85 and 125 kD respectively), even small aggregates could impair nuclear import and contribute to the loss of nuclear enrichment in WW domain mutants. However, if we could artificially drive nuclear localization of PASH-1::GFP and mCherry::DRSH-1 in G179R and W180A mutants, this would suggest that aggregation alone does not underlie the loss of nuclear enrichment. To test this, we added sequences encoding EGL-13 and SV40 nuclear localization signals (NLSs) onto either end of *GFP* in *pash-1::GFP* and *mCherry* in *mCherry::drsh-1* in wild-type and WW domain mutant backgrounds, as these two NLSs cause strong nuclear enrichment of *C. elegans* proteins[41]. PASH-1::GFP and mCherry::DRSH-1

containing the two ectopic NLSs did not show enhanced nuclear enrichment in wild-type backgrounds (Supplementary Fig. 5b, c). Nevertheless, the PASH-1 WW domain mutants with the ectopic NLSs displayed a partial restoration of PASH-1 nuclear enrichment. (Supplementary Fig. 5b, c). Similarly, mCherry::DRSH-1 with the ectopic NLSs also displayed partial rescue of nuclear enrichment in the mutant backgrounds; however, with both proteins, rescue was variable and never complete (Supplementary Fig. 5b, c). While these results do not rule out aggregation, they suggest that the cytoplasmic fractions of PASH-1 and DRSH-1 in the WW domain mutants retain the capacity to localize to the nucleus. This implies that protein aggregation is not solely responsible for the loss of nuclear enrichment in these mutants. It remains possible that PASH-1 misfolds in these mutants, potentially causing misfolding of DRSH-1 as well, however, we did not investigate this possibility further.

The loss of nuclear enrichment of DRSH-1 in *pash-1* G179R and W180A mutants suggests a fundamental role for PASH-1 in facilitating proper nuclear localization of DRSH-1. This led us to explore whether DRSH-1 also has a role in promoting nuclear localization of PASH-1. We found that RNAi-knockdown of *drsh-1* caused PASH-1::GFP to lose its nuclear enrichment (Fig. 5b and Supplementary Fig. 5d, e). Likewise, depletion of *pash-1* by RNAi led to a reduction in mCherry::DRSH-1 nuclear enrichment similar to what we observed in WW domain mutants (Fig. 5b and Supplementary Fig. 5d, e). Therefore, PASH-1 and DRSH-1 are required for each other's nuclear localization, but whether the mechanism relates to a cooperative role in nuclear entry or retention or to a role in proper protein folding is unclear.

We next tested whether the two ectopic NLSs introduced into PASH-1::GFP and mCherry::DRSH-1 could rescue their nuclear localization following reciprocal RNAi knockdown of *drsh-1* and *pash-1*. The ectopic NLSs on mCherry::DRSH-1 partially restored its nuclear enrichment following *pash-1* RNAi, and the ectopic NLSs on PASH-1::GFP similarly led to partial rescue of its nuclear enrichment following *drsh-1* RNAi (Supplementary Fig. 5f). These results indicate that DRSH-1 and PASH-1 can still localize to the nucleus in each other's absence when nuclear transport is artificially enhanced. That nuclear localization was only partially restored by the ectopic NLSs underscores the importance of proper Microprocessor formation in promoting nuclear entry or retention of PASH-1 and DRSH-1.

In mammals, the Microprocessor regulates each of its core constituents, DGCR8 and DROSHA[21]. DGCR8 is downregulated through Microprocessor cleavage of two hairpins in its mRNA, while DROSHA is stabilized through protein-protein interactions involving DGCR8[21,42]. We took advantage of the *pash-1* and *drsh-1* RNAi-knockdown assays described above to assess whether crosstalk also occurs in *C. elegans*. RNAi-knockdown of *pash-1* or *drsh-1* reduced their mRNA levels by ~75% and ~65%, respectively, and led to a similar decrease in PASH-1::GFP and mCherry::DRSH-1 protein levels, indicating that the RNAi treatment was moderately effective (Fig. 5c, d). *drsh-1* knockdown coincided with an ~25% increase in *pash-1::GFP* mRNA levels but had no discernable impact on PASH-1::GFP protein levels (Fig. 5c, d). RNAi knockdown of *pash-1* did not significantly affect *drsh-1* mRNA or protein levels, suggesting that *drsh-1* expression is not regulated by the Microprocessor despite its nuclear localization being dependent on PASH-1 (Fig. 5c, d). In *C. elegans*, the *pash-1* mRNA lacks the hairpins found in other species[21]. Nevertheless, it could still be indirectly regulated by the Microprocessor, as it is predicted to be a target of miR-71[43]. Consistent with this possibility, we previously observed a modest increase in *pash-1* mRNA levels in the absence of the major miRNA Argonaute *alg-1*[44]. However, we did not detect an increase in *pash-1* mRNA levels in *mir-71* deletion mutants (Supplementary Fig. 5g). Furthermore, PASH-1::GFP protein levels were not upregulated when we scrambled the *mir-71* binding site in the *pash-1* 3' UTR (Supplementary Fig. 5h).

We did not detect a discernable increase in PASH-1::GFP or mCherry::DRSH-1 expression in embryos from animals treated with RNAi against the major miRNA Argonautes, *alg-1* and *alg-2*, although modest differences would be difficult to detect by fluorescence and we did not explore this further (Supplementary Fig. 5i). Importantly, however, *alg-1* and *alg-2* knockdown did not affect PASH-1 and DRSH-1 localization to the nucleus, indicating that miRNAs are unlikely to regulate subcellular localization of the Microprocessor (Supplementary Fig. 5i). These results suggest that any crossregulation that might exist between PASH-1 and DRSH-1 in *C. elegans* is not likely to have a major role in their protein expression levels. However, given that both proteins depend on each other for their nuclear localization, they nevertheless share an important functional regulatory interaction.

## PASH-1's WW domain promotes microprocessor assembly

It is possible that the loss of nuclear localization of PASH-1 and DRSH-1 in the WW domain mutants is due to misassembly of the Microprocessor. To test this, we first did size-exclusion chromatography on animals expressing mCherry::DRSH-1 and either wild-type or the G179R or W180A mutant form of PASH-1 fused to GFP. We identified several fractions ranging from ~158 to 669+ kDa that contained both DRSH-1 and PASH-1 (Fig. 6a). The Western blot signals for PASH-1::GFP and mCherry:DRSH-1 were significantly reduced in the G179R and W180A mutants. This reduction may be due to the previously noted lower levels of mutant PASH-1::GFP, as well as decreased protein solubility in these mutants, which was observed in WW domain mutants of human and fly DGCR8/Pasha, however, we did not normalize by total protein levels in these experiments[16,24]. Despite differences in protein levels, large complexes still formed in both G179R and W180A mutants, suggesting that these residues are not totally essential for complex formation. (Fig. 6a). It is noteworthy that a substantial proportion of DRSH-1 migrated in the size range consistent with a single molecule of mCherry::DRSH-1 (~150 kDa), even in the strain containing unaltered PASH-1::GFP, which may be indicative of a role for DRSH-1 apart from the Microprocessor (Fig. 6a). We did not observe large aggregates in the void fractions, consistent with the proteins not being aggregated (Fig. 6a). However, we observed a shift in the elution profile of mCherry::DRSH-1 toward higher molecular weight fractions in *pash-1*[W180A]::*GFP* mutants (Fig. 6a). While this shift could suggest a tendency for small-scale oligomerization or conformational changes, the absence of a similar shift in *pash-1*[G179R]::*GFP* mutants indicates that it is unlikely to be the underlying cause of mislocalization in these mutants (Fig. 6a).

To further explore assembly of the Microprocessor in PASH-1 WW domain mutants, we did reciprocal co-IPs of PASH-1 and DRSH-1 from animals containing wild-type or the G179R or W180A mutant forms of PASH-1 fused to GFP and wild-type DRSH-1 fused to mCherry. Wild-type PASH-1::GFP co-IP'd with mCherry::DRSH-1 effectively, but this interaction was significantly diminished in PASH-1 G179R mutants (Fig. 6b and Supplementary Fig. 6a, b). Similarly, mCherry::DRSH-1 co-IP'd more efficiently with non-mutant PASH-1::GFP than with the G179R mutant form (Supplementary Fig. 6a, b). Therefore, the G179 residue contributes to the assembly or stability of the Microprocessor. We also observed a trend of reduced interaction between PASH-1::GFP and mCherry::DRSH-1 in the W180A form of PASH-1. However, due to high variability among biological replicates, possibly resulting from developmental defects in *pash-1*[W180A] mutant animals, this difference was not statistically significant (Fig. 6c and Supplementary Fig. 6c, d).

Interestingly, an expanded region of PASH-1 encompassing the WW domain is sufficient for dimerization in vitro[17]. Hence, to further test the involvement of G179 and W180 in Microprocessor assembly, we explored the idea that these residues promote PASH-1 dimerization, which could in turn explain the loss of Microprocessor integrity in these mutants. To do so, we fused sequence encoding a 3xFLAG peptide to the sequence encoding the PASH-1 dimerization region (amino acids 149-266), expressed it under the control of ubiquitin (*ubl-1*) regulatory elements, and integrated it into the *C. elegans* genome as a single-copy transgene, *FLAG::pash-1*[149-266]. We then introduced it by mating into strains containing *mCherry::drsh-1* and either *pash-1::GFP* or the *pash-1*[G179R]::*GFP* or *pash-1*[W180A]::*GFP* mutants. Next, we co-IP'd FLAG::PASH-1[149-266] and tested for its interaction with the wild-type and mutant forms of PASH-1::GFP, as well as mCherry::DRSH-1. Wild-type PASH-1::GFP was readily detectable in FLAG::PASH-1[149-266] co-IPs, confirming that the dimerization domain identified in vitro interacts with full-length PASH-1 in vivo (Fig. 6d and Supplementary Fig. 6e, f). In contrast, PASH-1[G179R]::GFP and PASH-1[W180A]::GFP were not detectable above background in FLAG::PASH-1[149-266] co-IPs (Fig. 6d and Supplementary Fig. 6e, f). mCherry::DRSH-1 also co-IP'd with FLAG::PASH-1[149-266] in the

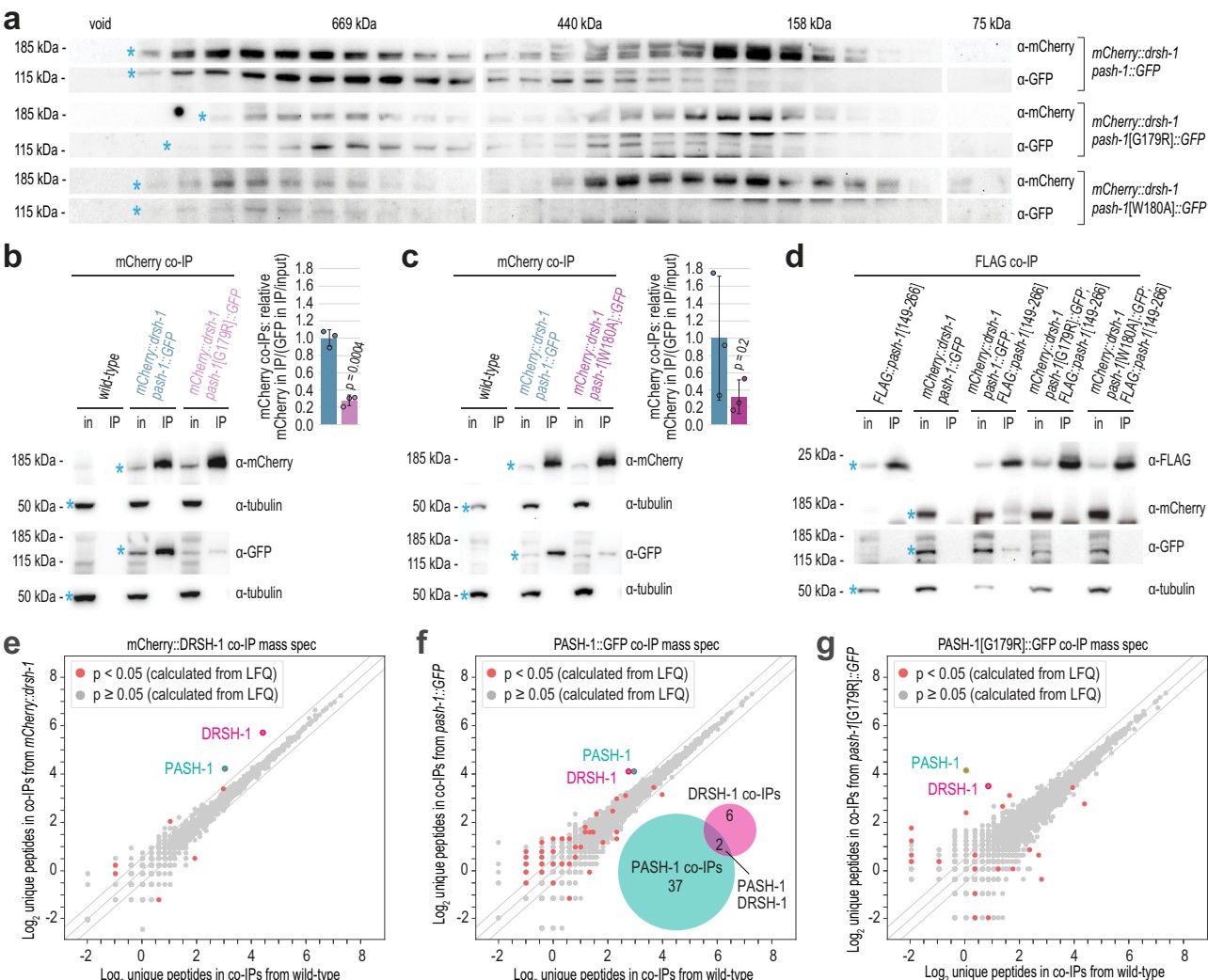

**Fig. 6 | Impaired Microprocessor assembly in PASH-1 WW domain mutants.**
**a** Western blot analysis of wild-type and mutant PASH-1::GFP and wild-type
mCherry::DRSH-1 from protein fractions captured with size exclusion chromato-
graphy. Masses are approximated based on size markers. The void fractions were
determined based on protein elution volumes and confirmed with high molecular
weight standards not run in parallel. Asterisks mark expected bands. **b**, **c** Western
blot analysis of PASH-1::GFP and PASH-1[G179R]::GFP (**b**) or PASH-1[W180A]::GFP (**c**)
co-IP'd with mCherry::DRSH-1. in, cell lysate input fraction; IP, co-IP fraction.
Tubulin is shown as a loading control. Asterisks mark expected bands. The bar plots
show the mean ratio of mCherry levels in IP fractions relative to GFP levels in IP
fractions normalized to GFP levels in input fractions. Blue bar: *mCherry::drsh-1;
pash-1::GFP*. Red bar: *mCherry::drsh-1; pash-1*[G179R]::*GFP*. Values are relative to the
*mCherry::drsh-1 pash-1::GFP* control. Two-tailed, two-sample Student's t-tests were
used to calculate the *p*-values. Error bars are SD. *n* = 3 biological replicates. Samples

derived from the same experiment and blots were processed in parallel. Blot
images for one of 3 replicates are shown (Supplementary Fig. 6a–d). **d** Western
blot analysis of PASH-1::GFP, PASH-1[G179R]::GFP, PASH-1[W180A]::GFP and mCher-
ry::DRSH-1 co-IP'd with FLAG::PASH-1[149-266]. Tubulin is shown as a loading
control. Asterisks mark expected bands. Blot images for one of 4 biological repli-
cates from 2 independent experiments are shown (Supplementary Fig. 6e–f).
**e–g** Mass spectrometry analysis of mCherry::DRSH-1 (**e**), PASH-1::GFP (**f**), and PASH-
1[G179R]::GFP (**g**) complexes from protein co-IPs. Scatter plots display the average
log$_2$ unique peptide counts in co-IPs from wild-type and the indicated transgenic
strains. *n* = 4 biological replicates for each strain. Diagonal lines show 0-, 2-, and -2-
fold enrichments. *p*-values were calculated based on label free quantification (LFQ)
using a modified t-statistic (see Methods). The inset Venn diagram in (**f**) shows the
overlap between mCherry::DRSH-1 and PASH-1::GFP interactors (*p* < 0.05). Source
data are provided as a Source Data file.

presence of non-mutant PASH-1::GFP, albeit at relatively low levels, but
not in the presence of the G179R or W180A mutants (Fig. 6d and
Supplementary Fig. 6e, f). These results demonstrate that the G179 and
W180 residues are important for PASH-1's dimerization and proper
assembly of the Microprocessor.

To identify the stable components of the Microprocessor and to
determine if the composition changes in complexes containing the
PASH-1[G179R] mutation, we did quantitative protein mass spectro-
metry. We did not identify any proteins that were enriched in both
mCherry::DRSH-1 and PASH-1::GFP co-IPs aside from DRSH-1 and PASH-
1 using either label free quantification (LFQ; Supplementary Data 2) or
unique peptide counts as a measurement (Fig. 6e, f). This suggests that

DRSH-1 and PASH-1 are the only major stable components of the
Microprocessor in *C. elegans*, although more transient interactors may
also exist. Proteins identified in co-IPs from only one of the strains,
*mCherry:drsh-1* or *pash-1::GFP*, could be background or unique to one
protein or the other and were not considered further. In DRSH-1 and
PASH-1 co-IPs from animals containing unaltered PASH-1::GFP we
observed similar enrichment of unique peptides for both DRSH-1 and
PASH-1 (Fig. 6e, f). In contrast, unique peptides of PASH-1 were enri-
ched nearly 2-fold over unique DRSH-1 peptides in PASH-1[G179R]::GFP
co-IPs, suggesting a weaker interaction between these proteins in the
G179R-mutant, consistent with the co-IP Western blot experiment
above (Fig. 6g).

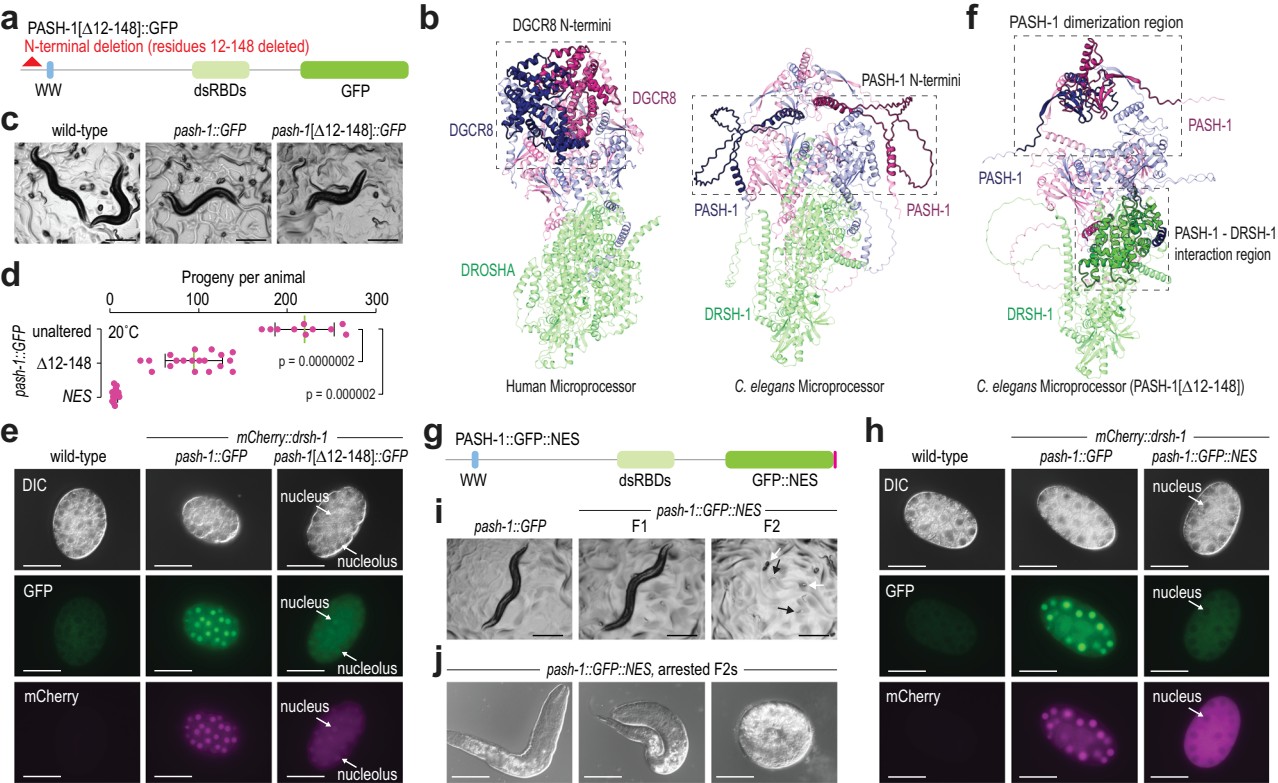

**Fig. 7 | Nuclear localization of the Microprocessor is required for development.**
**a** Diagram of the N-terminal PASH-1[Δ12-148]::GFP deletion protein with the WW domain (WW), double-stranded RNA-binding domains (dsRBDs), and GFP indicated. **b** AlphaFold3-predicted structures of the Microprocessor from humans (with only DROSHA and DGCR8) and *C. elegans*. The N-termini of human DGCR8 (residues 1-275) and *C. elegans* PASH-1 (residues 12-148) are highlighted.
**c** Representative images of wild-type, *mCherry::drsh-1 pash-1::GFP*, and *mCherry::drsh-1 pash-1[Δ12-148]::GFP* animals. 1–2 gravid adults, larvae, and eggs are visible. The scale bars are 0.3 mm. At least four representative adults for each strain were imaged. **d** Numbers of progeny produced by *pash-1::GFP* (unaltered), *pash-1[Δ12-148]::GFP* (Δ12-148), and *pash-1::GFP::NES* (NES) animals grown at 20 °C. Error bars are SD. $n = 10$ (unaltered), 19 (Δ12-148), or 14 (NES) animals. *p*-values were calculated using two-tailed Mann-Whitney U tests. **e** PASH-1::GFP, PASH-1[Δ12-148]::GFP, and mCherry::DRSH-1 expression in embryos. A wild-type embryo is shown as a control. The scale bars are 0.03 mm. At least 6 representative embryos for each strain were imaged. **f** AlphaFold3-predicted structure of the *C. elegans*

Microprocessor containing PASH-1[Δ12-148]. PASH-1's dimerization region (residues 148-266) and the PASH-1-DRSH-1 interacting region (PASH-1 helix [residues 497-513], DRSH-1 RIIIDa [residues 683-803], and DRSH-1 RIIIDb [residues 830-981]) are highlighted. **g** Diagram of the PASH-1::GFP protein fused to a nuclear export signal (NES, pink) with the WW domain (WW), dsRBDs, and GFP indicated. **h** PASH-1::GFP, PASH-1::GFP::NES, and mCherry::DRSH-1 expression in embryos. A wild-type embryo lacking GFP is shown as a control. The scale bars are 0.03 mm. At least six representative embryos for each strain were imaged. **i** Representative images of *pash-1::GFP* (control) and *pash-1::GFP::NES* animals. F1 animals are first-generation segregates from a heterozygous parent. F2 are descended from a homozygous parent. White arrows point to arrested embryos, and black arrows point to arrested larvae. The scale bars are 0.3 mm. At least 2 representative individuals for each strain were imaged. **j** Arrested *pash-1::GFP::NES* larvae and embryos. The scale bars are 0.03 mm. Three representative individuals were imaged. Source data are provided as a Source Data file.

## Microprocessor nuclear activity is critical for development

Given our results that *pash-1[G179R]::GFP* and *pash-1[W180A]::GFP* mutants are viable despite mislocalization of PASH-1 and DRSH-1 to the cytoplasm, nuclear localization of the Microprocessor may not be essential for *C. elegans* development. Human DGCR8 contains an NLS at its N-terminus[34]. Although this region is not structurally conserved in *C. elegans* and lacks an obvious NLS, we deleted it in the *mCherry::drsh-1 pash-1::GFP* strain to determine if we could prevent PASH-1's nuclear localization and thereby assess whether nuclear activity of the Microprocessor is required for development (Fig. 7a, b). Animals containing the N-terminal deletion (*pash-1[Δ12-148]::GFP*) were slightly dumpy and produced fewer viable progeny than non-mutant animals (Fig. 7c, d). However, because the N-terminal deletion removes 137 amino acids, it could disrupt other features of the PASH-1 protein, which may underly the fertility defects and dumpy phenotype. The deletion resulted in a modest but variable reduction in diffuse nuclear enrichment, which was often coincident with an increase in nucleolar enrichment (Fig. 7e and Supplementary Fig. 7a). mCherry::DRSH-1 colocalized consistently with PASH-1[Δ12-148]::GFP even in nucleoli (Fig. 7e and Supplementary Fig. 7a). The N-terminal deletion was not

predicted by AlphaFold3 to disrupt the interaction between PASH-1 and DRSH-1 (Fig. 7f). Furthermore, mCherry::DRSH-1 co-IP'd with PASH-1[Δ12-148]::GFP, indicating that the Microprocessor complex still forms in the mutant (Supplementary Fig. 7b).

Since the PASH-1[Δ12-148]::GFP mutant may lack an NLS important for nuclear localization of the Microprocessor, we tested whether introducing an ectopic NLS could rescue the nuclear localization of the mutant. Introducing a single EGL-13 NLS-encoding sequence onto the 5' end of GFP in *pash-1[Δ12-148]::GFP* caused both PASH-1[Δ12-148]::GFP and mCherry::DRSH-1 to localize even more strongly to nucleoli and further diminished non-nucleolar nuclear enrichment (Supplementary Fig. 7a). The ectopic NLS also led to a further reduction in the number of progeny and a significant increase in the occurrence of burst animals (Supplementary Fig. 7c, d). Our efforts to develop a strain with a second NLS-encoding sequence on *pash-1[Δ12-148]::NLS::GFP* were unsuccessful, possibly because of lethality. It is unclear if PASH-1's N-terminus has a direct role in preventing nucleolar enrichment, or if instead, misfolding or aggregation of the truncated protein causes it to be sequestered in nucleoli. However, that the N-terminal deletion still colocalized and co-IP'd with mCherry::DRSH-1

indicates that the proteins still form a complex. It is possible that the complex functions aberrantly within the nucleolus, which could underlie the reduced fertility and bursting we observed.

Because PASH-1 still partially localized to the nucleus in the N-terminal deletion mutant, it did not allow us to determine if nuclear localization is essential for development as we had set out to do. Therefore, we added a nuclear export signal (NES)-encoding sequence to the 3′ end of *pash-1::GFP* (*pash-1::GFP::NES*) and introduced it by mating into animals containing *mCherry::drsh-1* (Fig. 7g)[45,46]. The ectopic NES on PASH-1::GFP drove both PASH-1::GFP and mCherry::DRSH-1 to strongly mislocalize to the cytoplasm, demonstrating that DRSH-1 does not dissociate from PASH-1 and enter or remain in the nucleus outside of the Microprocessor complex (Fig. 7h). This mutant allowed us to investigate the necessity of nuclear localization for the Microprocessor. *pash-1::GFP::NES* mutants displayed a high incidence of embryonic arrest and, when hatched, arrested early in development. However, like other *pash-1* mutant alleles, homozygous mutant F1 progeny of heterozygous animals appeared normal (Fig. 7i, j). These results underscore the critical requirement of nuclear localization for Microprocessor function.

## Discussion

We uncovered a cross-regulatory interaction between PASH-1 and DRSH-1 important for their localization to the nucleus, which may be mediated by PASH-1 dimerization via the GW motif in its WW domain. Our findings suggest that entry or retention of PASH-1 and DRSH-1 in the nucleus is dependent on association with an intact Microprocessor complex. This was unexpected because both proteins seem to have their own nuclear localization signals in mammals[34,35]. However, the N-terminal regions containing these signals are poorly conserved in *C. elegans*. Nevertheless, the loss of nuclear enrichment when we deleted the N-terminus of PASH-1 suggests that it may have a cryptic NLS. Introducing an ectopic NLS did not rescue diffuse nuclear localization of the N-terminal deletion mutant but instead enhanced its nucleolar localization, suggesting that the N-terminus has another role in PASH-1 function.

Protein modifications may be important for nuclear localization of the Microprocessor and such modification could depend on an intact complex. In humans, phosphorylation of DROSHA promotes its nuclear localization[35]. Such regulation may limit Microprocessor-independent functions of DGCR8/Pasha or Drosha in the nucleus, which could disrupt gene regulation if not tightly controlled. Pasha has been shown to regulate neuronal genes in the absence of Drosha in *Drosophila* and DGCR8 regulates snoRNAs independent of DROSHA in human cells[47,48]. Furthermore, our protein fractionation experiments suggest that a substantial fraction of DRSH-1 is not in complex with PASH-1, further pointing to the possibility of independent functions for these proteins. Thus, there is a precedent for the individual Microprocessor components to function independently of each other, which could have unintended consequences if not tightly regulated. While we did not identify a post-transcriptional gene regulatory mechanism between PASH-1 and DRSH-1 like that of mammals[21], cross-regulation of protein localization may serve a similar role in maintaining their equilibrium in the nucleus.

Despite a crucial role for PASH-1 in *C. elegans* development, mutations within its WW domain are relatively well tolerated. While the W180A mutation caused severe developmental defects and reduced fertility, the neighboring G179R mutation caused only mild defects and near-normal fertility, even though both residues are similarly required for DRSH-1 interaction and nuclear localization. This suggests that Microprocessor assembly and nuclear localization defects only partially explain the W180A phenotype. The W180 residue may have additional roles, such as facilitating transient interactions important

for miRNA processing or in proper protein folding. These roles may involve heme binding or protein orientation on miRNA hairpins, but further investigation is needed. Since our mass spectrometry analysis did not detect additional protein components of the Microprocessor, additional interactions may be weak or transient.

Pri-miRNA processing is well-established to occur in the nucleus[8,33,36]. In humans, processing is thought to be largely co-transcriptional, facilitated by DROSHA's N-terminal proline-rich region[49–51]. However, this region is absent in *C. elegans*[51]. It is possible that, unlike in humans, *C. elegans* miRNAs are processed after transcription and splicing are completed. While most miRNAs occur in the introns of protein-coding or non-coding genes in humans, only a small subset are produced from presumptive introns in *C. elegans*. Even the introns that contain miRNAs in *C. elegans* appear to have regulatory elements distinct from the host transcript and thus may not undergo excision, which could be important for pri-miRNA processing if it occurs after transcription, since introns are typically degraded during splicing[52,53]. Additionally, introducing an intron into a miRNA hairpin does not disrupt processing of the hairpin or alter the mature miRNA sequence in *C. elegans*, indicating that splicing occurs before pri-miRNA processing[54]. In contrast, in humans, pri-miRNAs spanning intron-exon junctions are largely excluded from processing by the Microprocessor[55]. Regardless of timing, our results demonstrate that excluding the Microprocessor from the nucleus leads to embryonic or early larval lethality, emphasizing a critical requirement for nuclear processing.

## Methods
### Strains

To generate the pri-miR-58 sensor, TAM109[*ram3([pCMP1]ubl-1::mCherry::pri-mir-58 + Cbr-unc-119( + ))* II], 3,000 base pairs (bp) upstream of the *ubl-1* start codon was amplified from wild-type *C. elegans* (N2) genomic DNA using Phusion polymerase (ThermoFisher, F534L) and cloned into pDONR 221 P1-P4 (Addgene plasmid #186351) using Gateway Recombination Cloning Technology (ThermoFisher) following the manufacturer's protocol[56]. mCherry was PCR amplified from plasmid DNA with the primers GGGGACAACTTTTCTATACAAAGTTGACATGGT CTCAAAGGGTGAAGAAG and GGGGACAACTTTATTATACAAAGTTGT TTACTTATACAATTCATCCATG and cloned into pDONR 221 P4r-P3r (Addgene plasmid #121527). The miR-58 hairpin sequence and 100 bp upstream and 1,000 bp downstream were amplified from wild-type genomic DNA and cloned into pDONR 221 P3-P2 (Thermo Fisher). The entry plasmids were recombined into the destination vector CMP1 and then transformed into *C. elegans* strain EG6699 using Mos1-mediated single copy insertion[20,56]. VC1138[*drsh-1(ok369) I/hT2[bli-4(e937) let-?(q782) qIs48]*(I;III)], MT15024[*mir-58(n4640)* IV], MT12993[*mir-71(n4115)* I], and KW2090[*cdk-9(tm2884) I/hT2 [bli-4(e937) let-?(q782) qIs48]* (I;III)] were previously described[32,38,57]. TAM93[*pash-1(ram33)* I] containing the 38a mutation that was identified in a forward genetic screen was outcrossed three times to wild-type (N2). PHX4329[*pash-1(syb4327)* I] (*pash-1*[W180A]), PHX6091[*pash-1(syb6091)* I] (*pash-1::GFP*), PHX6157[*pash-1(ram33 syb6157)* I] (*pash-1*[G179R]::*GFP*), PHX6655[*pash-1(syb6091 syb6655)* I] (*pash-1*[W180A]::*GFP*), PHX6628[*drsh-1(syb6628)* I] (*mCherry::drsh-1*), PHX7689[*pash-1(syb66091 syb7689)* I] (*pash-1*[△12-148]::*GFP*), PHX8549[*pash-1(syb6091 syb8549)* I] (*pash-1*[GW-mut]::*GFP*), PHX8348 [*pash-1(syb8348)/hT2[bli-4(e937) let-?(q782) qIs48]* (I;III)] (*pash-1::GFP::NES*, containing NES sequence encoding LALKLAGLDI), and PHX6604[*pash-1(syb6091 syb6604)* I] (*pash-1*[miR-71mut]::*GFP*, miR-71 seed target edited from TCTTTCA to AGAAAGT) were generated by Suny Biotechnology (Fuzhou, China) using CRISPR-Cas9 genome editing and outcrossed to wild-type (N2) to remove potential background mutations or to each other to generate combinatorial strains (Supplementary Data 3)[25–28]. Additional strains are described in the Supplementary Information.

## Animal growth conditions

For the mass spectrometry experiment, animals were expanded as described below[58]. For all other experiments, animals were grown on NGM plates containing *E. coli* OP50 at 20 °C unless noted otherwise.

## Forward genetics

L4 and young adult *C. elegans* were rotated in M9 buffer solution containing 1% ethyl methyl sulfonate (EMS) solution for 4 h at room temperature and then washed 4x in M9. For each experiment, 50 mutagenized P0 animals were plated onto 20 10 cm NGM plates seeded with *E. coli* OP50. In the F1 screen for cis-acting factors (which identified the pri-miR-58 sensor mutation), 500 bright fluorescent animals were selected from among ~100,000 F1 animals. In the F2 screen (which identified the *pash-1(ram33*[G179R]*)* mutation), up to 4 bright fluorescent individuals were selected from each of 20 distinct pools of F2 animals, each derived from ~2000 F1s. For the F1 screen, individual F1 animals were lysed, and the miR-58 hairpin sequence of the pri-miR-58 sensor was sequenced using Sanger sequencing. For the F2 screen, individual animals were isolated and then expanded for several generations. The brightest healthy line after several generations was subjected to whole genome sequencing to identify the causal mutation.

## Genome sequencing

DNA was isolated from EMS line 38a using the Gentra Puregene Tissue Kit following the manufacturer's protocol (QIAGEN, 158667). The genome sequencing library was prepared and sequenced using Novogene's Whole Genome Sequencing service, which utilizes the NEBNext DNA Library Prep Kit, and sequenced on an Illumina NovaSeq (PE150). Data was processed and single nucleotide polymorphisms (SNPs) were identified using Novogene's bioinformatics analysis pipeline. Briefly, reads were aligned to the *C. elegans* genome (WS190) using BWA (parameters: 'mem -t 4 -k 32 -M')[59]. PCR duplicates were removed using SAMTOOLS[60]. SNPs were called using GATK (parameters: '-T HaplotypeCaller --gcpHMM 10 -stand_emit_conf 10 -stand_call_conf 30')[61]. Results were filtered to reduce the error rate (parameters: 'QD < 2.0 || FS > 60.0 || MQ < 30.0 || HaplotypeScore > 13.0 || MappingQualityRankSum < -12.5 || ReadPosRankSum < -8.0'). SNPs were annotated using ANNOVAR[62].

## RNA and protein structure analysis

Secondary structures of miRNA hairpins were predicted with RNAfold[63]. Structures of human and *C. elegans* Microprocessor complexes were predicted using AlphaFold3 with protein sequences obtained from UniProt[64,65]. The X-ray diffraction structure of the human DGCR8 WW domain (PDB: 3LE4) was described by Senturia et al.[17]. The structures were analyzed and images generated using ChimeraX[66].

## RNA isolation

Gravid adult animals, grown for 72 h at 20 °C or 54 h at 25 °C after L1 synchronization, were washed off plates and then washed 3x in M9 buffer and flash frozen in liquid nitrogen. RNA was isolated from whole gravid adult animals using TRIzol (Life Technologies, cat# 15596018) according to the manufacturer's recommendations but with a second chloroform extraction step. For mRNA qRT-PCR, total RNA was DNase-treated using the TURBO DNA-free Kit (ThermoFisher, AM1907) according to the manufacturer's protocol.

## qRT-PCR

Small RNA qRT-PCR was done with TaqMan reagents and custom probes targeting miR-1 (Life Technologies, assay name: hsa-miR-1, 4427975), miR-58a-3p (sequence: UGAGAUCGUUCAGUACGG CAAU), miR-35-3p (sequence: UCACCGGGUGGAAACUAGCAGU), 21UR-1 (sequence: UGGUACGUACGUUAACCGUGC), and 22G-rRNA

(sequence: GAAGAAAACUCUAGCUCGGUCU) following the manufacturer's protocol (Life Technologies, 4331348). Pri-miR-58 qRT-PCR was done with custom TaqMan probe sets targeting *pri-miR-58* and *act-1*. Other pri-miRNAs were analyzed using SYBR Green and the following primer sets: pri-miR-35, CTCTCCTAATTTCCATTCCC and GTGGAGCA AGTGGAAAAGATC; pri-miR-51, ACCAACATTTGCCTGCTCAC and GTACCTGCTTCCATGTTCAC; pri-miR-80, CCAACTTTTTGGTGCTTA TTCC and ATACTTCAGGTTGTGAATGTGG; and pri-miR-238, GAAGGC AGTCTCTTTTGTCAG and ATCTGAATGGCATCGGAGTAC. *pash-1* and *drsh-1* mRNA qRT-PCR analysis was done using SYBR Green and the following primers: *pash-1*, GTTCACTCGTGTCGTCACTC and CGTT TTCGTGCAGCTCATCC; and *drsh-1*, GTACTTGGAATCGAAGGACC and AGATTAGCCAAAGCCAGCTC. For pri-miRNA and mRNA SYBR Green-based qRT-PCR, *rpl-32* levels were used for normalization (*rpl-32* primers: CATGAGTCCGACAGATACCG and ACGAAGCGGGTTCTTCT GTC). Ct values were captured using a CF96 Real-Time PCR Detection System (Bio-Rad) and averaged across 3-4 technical replicates for each of 3-4 biological replicates. The 2-ddCt method was used to calculate relative fold changes between conditions[67].

## Small RNA sequencing and data analysis

Small RNA sequencing libraries were prepared using the NEBNext Multiplex Small RNA Library Prep Set for Illumina following the manufacturer's protocol with exclusion of the initial optional size selection step and the 3' ligation step changed to 16 °C for 18 h to improve capture of methylated small RNAs (New England Biolabs, E7300S). -134–148 nt small RNA PCR amplicons, corresponding to -16–30 nt small RNAs, were size selected on a 10% polyacrylamide gel, transferred by electrophoresis (400 mA for 40 min) to DE81 chromatography paper, eluted at 70 °C for 20 min in the presence of 1 M NaCl, and precipitated at −80 °C overnight in the presence of 13 ug/ml glycogen and 67% EtOH. End-to-end data analysis, including adapter trimming and quality filtering with fastp and genome alignment with bowtie, was done using the default configuration in tinyRNA with the *C. elegans* genome WS279 release[68–72]. To identify iso-miRs, selection rules were defined within tinyRNA to capture reads with 5' ends aligning 1-3 nucleotides upstream or downstream of the 5' end of the annotated miRNA. Pre-miRNA reads were identified with a selection rule that specified partial overlap with annotated pre-miRNA sequences but excluded reads corresponding to the annotated miRNA or potential iso-miRs defined above, due to hierarchical assignment to these features (Supplementary Data 4).

## Microscopy

All imaging was done using an Axio Imager Z2 Microscope (Zeiss). Live animal imaging directly on growth media was done using a Plan-Apochromat 10x/0.45 M27 objective and DIC accessories. For mCherry and GFP imaging, animals were mounted on glass slides prepared with 1.5% Agarose pads. Larvae and adults were immobilized in 25 μM sodium azide. Zeiss 38 HE GFP shift free and 63 HE mRFP shift free filter sets were used in combination with a Plan-Apochromat 10x/0.45 M27 objective or Plan-Apochromat 63x/1.4 oil DIC M27 objective. A DIC Slider PA 63x/1.4 III HR was used for 63x DIC imaging. Images were acquired with an Axiocam 506 monochrome camera using the ZEN 2.3 pro software (Zeiss). Contrast and brightness were adjusted uniformly for all images within a single experiment in Adobe Photoshop or Fiji (ImageJ2 v.2.14.0). mCherry fluorescence signal quantification was done in Fiji[73]. A Discovery V8 Stereo Microscope (Zeiss) was used for selection of mutagenized animals.

## RNAi

For RNAi knockdown, synchronized L1 larvae were placed on RNAi plates containing IPTG and *E. coli* HT115 expressing dsRNA matching *cdk-9* (Ahringer or Vidal libraries, H25P06.2), *ama-1* (Ahringer library, F36A4.7), *alg-1/alg-2* (Ahringer library, F48F7.1), *pash-1* (Ahringer

library, T22A3.5), *drsh-1* (Ahringer library, F26E4.10), or empty vector (L4440, Addgene #1654) and grown at 20 °C[74,75]. Where applicable, gravid adults were collected for protein and RNA isolation after 72 h of treatment.

### Protein co-immunoprecipitation

PASH-1::GFP, PASH-1[G179R]::GFP, PASH-1[W180A]::GFP, FLAG::PASH-1[149-266], PASH-1[Δ12-148]::GFP, mCherry::DRSH-1, and RNA Polymerase II were co-IP'd from three biological replicate samples with ~12,000 gravid adult animals each. Animals were washed from plates and then 3x in M9 salt buffer, frozen in liquid nitrogen, and lysed in 1.2 ml 50 mM Tris-Cl, pH 7.4, 100 mM KCl, 2.5 mM MgCl2, 0.1% Igepal CA-630, and 1x Pierce Protease Inhibitor Tablets (Pierce Biotechnology, cat# 88266). Cell lysates were cleared for 10 min at 12,000 RCF at 4 °C. Cleared lysates were split into input and co-IP fractions. For GFP and mCherry co-IPs, cell lysates were incubated with 25 µL ChromoTek GFP-Trap Magnetic Agarose Beads (Proteintech, gtma-100) (GFP co-IPs) or ChromoTek RFP-Trap Magnetic Agarose Beads (Proteintech, rtma-20) (mCherry co-IPs) for 1 h at 4 °C. For FLAG co-IPs, lysates were incubated with 7 µg FLAG antibody (Sigma, F3165) for 1 h at 4 °C. After 30 min of FLAG antibody incubation, 60 µL of Protein A agarose bead slurry (Roche, PROTAA-RO) was added. For Pol II-PASH-1::GFP co-IPs, Pol II was co-IP'd with 4 µg Pol II antibody (Santa Cruz Biotechnology, sc-537117) and PASH-1::GFP was co-IP'd with 5 µg GFP antibody (Invitrogen, A11120) for 90 min at 4 °C. After 45 min of antibody incubation, 50 µL of SureBeads Protein G Magnetic Agarose Bead slurry (Bio-Rad, 1614023) was added. Beads were washed three times in lysis buffer and protein was eluted at 95 °C for 5 min in 1x Blue Protein Loading Dye (New England Biolabs, B7703S).

### Size-exclusion chromatography

Animals were lysed as described above for co-IPs. Cell lysates were treated with 0.5 µg/µL RNase I for 1 h at 4 °C. Proteins were separated on a Superdex 200 Increase small-scale SEC column, 10/300 GL (Cytiva, 28990944) according to the manufacturer's protocol. Western blot analysis of the fractions was done as described below.

### Western blots

Co-IP'd proteins, cell lysates, and SEC fractions were resolved on Bolt 15-well 4-12% Bis-Tris Plus Gels (Life Technologies, NW04125BOX), transferred to nitrocellulose membranes (Life Technologies, LC2001), and probed with GFP (Santa Cruz Biotechnology, sc-9996 HRP, clone B-2, 1:100 dilution), mCherry (Proteintech, clone 6G6, 1:200 dilution), FLAG (Pierce, PA1-984B, RRID AB_347227, 1:500 dilution), Pol II (Santa Cruz Biotechnology, sc-537117, clone 8WG16, 1:500 dilution), tubulin (Abcam, ab40742, clone DM1A, RRID AB_880625, 1:1000 dilution), and actin (Abcam, ab3280, clone ACTN05 (C4), RRID AB_303668, 1:1000 dilution) antibodies. Blots were imaged and quantified on a FluorChem E Imaging System (ProteinSimple).

### Mass spectrometry

**Sample preparation.** Wild-type control animals and strains expressing PASH-1::GFP, PASH-1[G179R]::GFP, or mCherry::DRSH-1 were washed from NGM growth medium and treated with a hypochlorite solution (2% NaClO, 666 mM NaOH) to isolate embryos, which were then plated onto high-density egg plates, grown until the gravid adult stage, and hypochlorite-treated again. The embryos were hatched in M9 buffer (22 mM KH2PO4, 42 mM Na2HPO4, 85 mM NaCl, 1 mM MgSO4) and arrested L1 stage animals were plated on standard NGM plates containing *E. coli* OP50. Gravid adult animals (200 ul packed worms per replicate, 4 biological replicates per sample) were washed from NGM plates and then 3x in M9. Animals were flash frozen in liquid nitrogen and stored at -80 °C. They were then thawed and lysed using a Bioruptor Plus (Diagenode) sonicator (10 cycles of 30 sec ON, 30 sec OFF, high efficiency) in lysis buffer (25 mM Tris HCl pH 7.5, 150 mM NaCl,

1.5 mM MgCl2, 1 mM DTT, 0.1% Triton X-100, and 1 tablet/40 ml cOmplete Mini, EDTA-free protease Inhibitor cocktail, Roche, 11836170001). Cell lysates were cleared for 10 min at 21,000 RCF at 4 °C. BCA assays (Thermo Fisher Scientific, 23225) were used to normalize the starting protein amount. 20 µL ChromoTek GFP-Trap Magnetic Agarose Beads (Proteintech, gtma-100) or ChromoTek RFP-Trap Magnetic Agarose Beads (Proteintech, rtma-20) were washed 3x in 500 µL wash buffer (25 mM Tris HCL pH 7.5, 150 mM NaCl, 1.5 mM MgCl2, and 1 mM DTT) and then incubated with cleared cell lysates overnight while rotating at 4 °C. Beads were washed 3x in wash buffer, resuspended in 50 µL 1x NuPAGE LDS Sample Buffer (Invitrogen, NP0007) and incubated at 95 °C for 10 min to elute protein.

**Enzymatic protein digestion.** Eluted proteins were separated briefly on a 10% NuPAGE Bis-Tris gel (Invitrogen, NP0301BOX), stained with Coomassie blue and cut into small gel cubes, followed by destaining in 50% ethanol/25 mM ammonium bicarbonate. The proteins were then reduced in 10 mM DTT at 56 °C and alkylated by 50 mM iodoacetamide in the dark at room temperature. Proteins were then digested by trypsin in 50 mM ammonium bicarbonate buffer overnight at 37 °C. Following peptide extraction through sequential incubation of the gel cubes in 30% and 100% acetonitrile, the sample volume was reduced in a centrifugal evaporator to remove residual acetonitrile. The resultant peptide solution was purified by solid phase extraction in C18 StageTips[76].

**Liquid chromatography tandem mass spectrometry.** Peptides were separated in an in-house packed 30 cm analytical column (inner diameter: 75 µm; ReproSil-Pur 120 C18-AQ 1.9-µm beads, Dr. Maisch GmbH) by online reverse phase chromatography through a 105 min non-linear gradient of 1.6–32% acetonitrile with 0.1% formic acid at a nanoflow rate of 225 nl/min. The eluted peptides were sprayed directly by electrospray ionization into a Q Exactive Plus Orbitrap mass spectrometer (Thermo Scientific). Mass spectrometry measurements were conducted in data-dependent acquisition mode using a top10 method with one full scan (scan range: 300–1650 m/z; resolution: 70,000, target value: $3 \times 10^6$, maximum injection time: 20 ms) followed by 10 fragmentation scans via higher energy collision dissociation (HCD; normalized collision energy: 25%, resolution: 17,500, target value: $1 \times 10^5$, maximum injection time: 120 ms, isolation window: 1.8 m/z). Precursor ions of unassigned or +1 charge state were rejected. Additionally, precursor ions already isolated for fragmentation were dynamically excluded for 20 s.

**Data processing and statistical analysis.** Mass spectrometry raw data were processed with MaxQuant (version 2.1.3.0)[77] using its built-in Andromeda search engine[78]. Spectral data were searched against a target-decoy database consisting of the forward and reverse sequences of the bait proteins, the UniProt *C. elegans* (release 2023_02; 28,540 entries) and *E. coli* (release 2023_01; 5,064 entries) reference proteomes, and a list of common contaminants[64]. Trypsin/P specificity was assigned. Carbamidomethylation of cysteine was set as fixed modification. Methionine oxidation and protein N-terminal acetylation were chosen as variable modifications. A maximum of 2 missed cleavages was tolerated. The "second peptides" options were switched on. The "match between runs" function was activated. The minimum peptide length was set to 7 amino acids. False discovery rate (FDR) was set to 1% at both peptide and protein levels.

The MaxLFQ algorithm[79] was employed for label-free protein quantification without using its default normalization option. Minimum LFQ ratio count was set to one. Both the unique and razor peptides were used for quantification. Detected *E. coli* proteins, reverse hits, potential contaminants and "only identified by site" protein groups were filtered out. Data normalization was performed on the log-transformed data. Under the assumption that the majority of the

detected proteins were non-specific background binders, two approaches were used: 1) For the DRSH-1 data, the distribution of protein intensities was assessed using kernel density estimation to find the peak density. The protein intensities were then normalized by adjusting the point at which peak density was found to the same value across all samples. 2) For the PASH-1 data, median-centering was performed based on the top 200 most abundant proteins detected in the control wild-type (N2) pull-downs. Proteins were further filtered to retain only those detected in at least two out of the 4 replicates in either group of each comparison. Following imputation of the missing LFQ intensity values, the statistical significance of the difference between the two groups was assessed using a modified t-statistic (t(SAM, statistical analysis of microarrays)[80]. The combined significance threshold (hyperbolic curve) was defined as $t0 = 1.3$–$1.4$ and $s0 = 1.5$.

## Fertility assays

Individual animals were grown from L1 or L2 stage larvae on OP50 at 20 °C or 25 °C, and their progeny were summed every day until the cessation of egg laying. Progeny counts included all larval stage animals.

## Statistics and reproducibility

For fertility assays, p-values were calculated using the two-tailed Mann-Whitney U test in GraphPad Prism. For qRT-PCR, bursting, and Western blot data analysis and in comparing total iso-miR and pre-miRNA read levels, two-tailed, two-sample Student's t-tests were used to calculate p-values in Microsoft Excel or GraphPad Prism. Whenever more than one comparison was made, p-values were adjusted using the Bonferroni method. DESeq2 was used within the tinyRNA sRNA-seq data analysis workflow to perform statistical analysis using the Wald test[68,81].

## Strain availability

All *C. elegans* strains used in this study are available from the Caenorhabditis Genetics Center (CGC) or upon request from the corresponding author.

## Data availability

Raw and processed high-throughput sequencing data generated in this study are available from the NCBI Gene Expression Omnibus (GEO) under accession number GSE263914. The mass spectrometry proteomics data have been deposited to the ProteomeXchange Consortium via the PRIDE partner repository with the dataset identifier PXD052084[82]. Source data are provided with this paper.

## Code availability

The tinyRNA v1.5.0 software used for analysis of sRNA-seq data is freely available from https://github.com/MontgomeryLab/tinyRNA.

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

## Acknowledgements

Jiaxuan Chen is gratefully acknowledged for his help with mass spectrometry and data analysis (instrumentation funded by the German Research Foundation, DFG INST 247/766-1 FUGG). Thanks also to Rachel Doser for help with strain generation, David Fay for guidance with AlphaFold and ChimeraX, and Alivia Ball and Maritza Soto-Ojeda for help with media and solutions. Strains not generated in this study were provided by the CGC, which is funded by the National Institutes of Health Office of Research Infrastructure Programs (P40 OD010440). The VC1138 strain was generated by the *C. elegans* Reverse Genetics Core Facility at the University of British Columbia, which is part of the international *C. elegans* Gene Knockout Consortium. This work was supported by the National Institutes of Health [R35GM119775 to T.A.M.], the Institute of Molecular Biology - Meinz [core funding to R.F.K.], and the German Research Foundation [DFG grant 252386272 to R.F.K.].

## Author contributions

T.L.K. and T.A.M. designed and planned experiments. T.L.K., B.E.M., K.J.R., M.C.C., E.R.C., M.J.S., R.A.S., C.N.M., and T.A.M. performed experiments. T.A.M. performed the sRNA-seq data analysis. I.J.I. and R.F.K. did the protein mass spectrometry experiments. H.S. assisted with size-exclusion chromatography. T.L.K. and T.A.M. wrote the manuscript. T.L.K., R.F.K., and T.A.M. edited the manuscript. T.A.M. acquired funding.

## Competing interests

The authors declare no competing interests.
