## [Transparent Peer Review file · Nature Communications]

Interdependence of Pasha and Drosha for localization and function of the Microprocessor in *C. elegans*

Corresponding Author: Dr Taiowa Montgomery

Version 0:

Reviewer comments:

Reviewer #1

(Remarks to the Author)

Montgomery et al. established a pri-miRNA processing reporter system in *C. elegans* and found that a G179R mutation in the WW domain of Pasha resulted in a pri-miRNA processing defect. The authors confirmed that the effect of the G179R mutation was similar to that of the W180A mutation, which is important in protein folding and dimerization of Pasha. Lastly, the authors claimed that the WW domain is required for the nuclear localization of the Drosha/Pasha complex.

Major points

(1) Because the WW domain mutations affect the protein folding and dimerization of Pasha, it is not convincing that the WW domain plays a specific role in the subcellular localization of Drosha/Pasha. I interpret the results as the accumulation of soluble aggregated Drosha and Pasha proteins in the cytoplasm where translation occurs.

It is known that the hydrophobic interaction between the second RNase III domain of human DROSHA and the C-terminal tail of DGCR8 is critical for DROSHA stability (Cell 164:81, 2016; ref. 42). Without this specific masking, human DROSHA protein becomes soluble aggregates (Cell 161:1374, 2015; ref. 14). I superimposed the AlphaFold2 predicted structure of *C. elegans* Drosha on the crystal structure of human DROSHA (PDB: 5B16). The second RNase III domain of *C. elegans* Drosha preserves surface hydrophobic residues (e.g., L920 and I967, corresponding to L1194 and V1243 of human DROSHA, respectively). Therefore, it seems that *C. elegans* Drosha would be aggregated (but still soluble) without the functional Pasha protein.

Then, I superimposed the AlphaFold2 predicted structure of *C. elegans* Pasha on the crystal structure of the human DGCR8 dimerization domain (PDB: 3LE4). G179R may affect the loop's flexibility and hinder the proper positioning of the side chain of F196. W180A will lose the critical hydrophobic dimerization interface. Therefore, the structural prediction implies that the WW domain mutations should affect Pasha's dimerization.

The authors showed that the WW domain mutations affected the protein folding and dimerization of Pasha and the assembly of the Microprocessor (Fig. 6B-D). Thus, without the need for a special nuclear localization mechanism through the WW domain, we can interpret the data through protein misfolding and aggregation in the cytoplasm.

mCherry-Drosha is enriched near 669 kDa and 158 kDa in the presence of wild-type Pasha, and both peak positions were shifted toward the left (larger size) in the W-mut animal, indicating *C. elegans* Drosha became aggregated without a properly folded/dimerized Pasha (Fig. 6A).

(2) The manuscript does not contain enough novel data to be published in Nat Commun. As the authors mentioned, the pri-miRNA processing reporter is not conceptually new (PNAS 111:1861, 2014; ref. 23). The processing deficiency of the cis-acting mutation found in the stem region of a pri-miRNA can be expected (Mol Cell 60:131, 2015). The trans-acting factor screening found only one Pasha mutant, whose molecular consequence can be interpreted with protein misfolding and aggregation.

Minor points

(1) Line 82. Because the pri-miRNA processing event occurs in the nucleus, I don't think the translational machinery affects this reduction.

- (2) Line 91. I think an internal loop is a more appropriate term than a bulge.
- (3) Line 148. The authors need to consider the kinetic difference between pri-miRNA and mature miRNA. A pri-miRNA is rapidly degraded because of DROSHA processing, but a mature miRNA is very stable.
- (4) Line 179. Because the G and W residues seem to work on the same function (i.e., dimerization), the term 'independent' may confuse readers.
- (5) Line 191. The authors can show the secondary structures of pri-miR-1829 and pri-miR-42 and explain why they are not optimal substrates for Microprocessor.
- (6) Fig 6A. The Pasha expression level of G- and W-mut are much lower than other Western blots. Are they all in void fractions?
- (7) Related to #6. Where are the void fractions? Based on Superdex 200's resolution, I think the first 3-5 fractions would be the void fractions.
- (8) Fig 4G. Protein levels of Drosha and Pasha need to be shown to exclude the possibility of other interpretations, such as transcriptional inhibition of the Drosha gene.
- (9) Line 300. There are two hairpins in the DGCR8 mRNA. One is in the 5' UTR, and another is in the CDS.
- (10) Line 371. typo, 'in in'
- (11) Line 450. The authors mentioned the possibility of other Drosha or other Pasha complexes. However, they had already done co-IP and MS experiments and could not find any other specific proteins.
- (12) Line 460 and 490. W180 may not have an additional role except for Pasha dimerization according to its structure (PDB: 3LE4) and MS experiment.

Reviewer #2

(Remarks to the Author)

Montgomery et al present findings related to the function of the WW domain of the Pasha (DGCR8) component of the microprocessor, which also includes the RNaseIII enzyme Drosha. The WW domain of Pasha had been previously shown to be necessary for Pasha dimerization, which is necessary for assembly of functional microprocessor — which consist of a Pasha dimer associated with a Drosha monomer. The authors show that mutations at two conserved amino acids of the *C. elegans* Pasha WW domain compromise microprocessor activity in vivo resulting in reduction of many microRNAs and consequent developmental phenotypes. Biochemical analysis of in vivo complexes, and fluorescence microscopy of tagged Drosha and Pasha in vivo, show that the WW domain mutations affect the nuclear localization of both the mutant Pasha protein and also Drosha.

Thus one of the central findings of the manuscript is a role for Pasha in facilitating Drosha's correct localization. Interestingly, the levels of expression of Pasha and Drosha reciprocally affect each others' nuclear localization. The authors show that depletion of Pasha by RNAi caused mCherry::DRSH-1 to accumulate primarily in the cytoplasm. Similarly, RNAi-knockdown of Drosha caused PASH-1::GFP to localize cytoplasmically. The authors tested whether the reciprocal mutual regulation of Pasha and Drosha reflects an indirect consequence of compromised microRNA activity, and found that RNAi knockdown of *alg-1* and *alg-2* did not affect Pasha and Drosha localization to the nucleus, consistent with a relatively direct mutual regulation.

A noteworthy contribution of the manuscript is the demonstration of a microprocessor sensor consisting of a transgene expressing a fluorescent protein (FP) from an mRNA containing a microRNA hairpin in its 3' UTR; normal microprocessor activity efficiently excises the hairpin from the reporter mRNA, destabilizing the mRNA and preventing FP expression; expression of the FP reveals compromised microprocessor activity on the reporter, enabling forward screens for mutations that compromise microprocessor activity. This reporter was used to isolate the WW mutation that serves as the basis for this study, which augurs well for the value of the reporter for future genetic studies of microprocessing in *C. elegans*.

Overall, the manuscript present important findings about the interdependence of microprocessor components in *C. elegans* for the proper assembly, cellular localization, and function of microprocessor. A specific significant contribution of the manuscript is the demonstration of the importance of certain amino acids in the Pasha WW domain for microprocessor assembly. A broader significant contribution is the implication of subunit stoichiometry in the dynamics of microprocessor cytoplasmic trafficking.

Minor critiques:

Line 81: "This reduction may be attributed to the translational machinery sequestering the transcript away from the miRNA pathway due to the presence of the mCherry coding sequence." This is confusing because the translational machinery would be in the cytoplasm, so it could not interfere with processing by microprocessor, if microprocessor is nuclear. Explain.

Line 93: "Heterozygous cis-acting mutations that disrupt cleavage of one copy of the sensor should partially de-silence mCherry expression, whereas only dominant or dosage-dependent heterozygous mutations in trans-acting factors would lead to de-silencing." This sentence is confusing as worded. It would be helpful to clarify that the context is the F1 self-progeny population from mutagenized hermaphrodites.

Line 107: "We draw two conclusions from these results. Firstly, the recognition or processing of the miR-58 hairpin is susceptible to slight perturbations in the miRNA duplex region. Secondly, the sensor accurately reflects the recognition and

processing of the miR-58 hairpin by the Microprocessor machinery.” The reporter itself is a valuable tool, but that is revealed by the recovery of the Pasha mutant. The cis-acting mutation in the reporter is not very informative, and raises more questions than it answers, such as why was it that only one out of 500 worms with de-silenced transgene contained a mutation. Also, the modest de-repression observed upon reconstructing that mutation is concerning. It would be better to remove from the narrative the presentation of the cis-screen.

147: “The lack of correlation we observed between primary and mature miR-58 suggests that the amount of pri-miR-58 exceeds what can be processed by the miRNA machinery.” The wording of this sentence is somewhat confusing. Perhaps the authors mean that microprocessing is not the rate-limiting biogenesis step in the accumulation of mature mir-58.

Line 176: “Surprisingly animals containing both mutations produced significantly fewer progeny than either of the single mutants” More explanation is required about why this is surprising.

Line 185: “At 25°C, a less ideal growth condition for *C. elegans*....” This is a rather vague statement; less ideal in what regard?

Line 192: “We did not confirm that the miRNAs identified in Ahmed et al. are authentic miRNAs, but it is possible that these are also non-Microprocessor dependent miRNAs as they were also upregulated.” Confusing. More explanation seems to be required. What is this statement in reference to? Mir-42 is a microRNA.

Line 195: “...mutants there is reduced competition for shared resources downstream of pri-miRNA processing due to the likely reduction in pre-miRNAs, which are subsequently diverted to mirtrons and other Microprocessor-independent miRNAs.” Addition explanation seems to be required to clarify that “upregulated” miRs could be those that are relatively less dependent than others on wt pasha.

Line 260: “...miR-58 sensor, which would implement it in pri-miRNA processing.” ‘implement’ should be “implicate”.

Line 455: “Interestingly, the W mutant is sick, whereas the G mutant is relatively healthy.” Please define ‘sick’ vs ‘healthy’. Are these qualitative difference or quantitative?

Line 471: “Therefore, it is possible that, unlike in mammals, *C. elegans* miRNAs are predominantly processed post-transcriptionally.” The definition ‘post-transcriptionally’ in this context is a bit ambiguous. Perhaps the salient distinction would be whether A) the microRNA hairpin is located in an intron of an mRNA such that microprocessing occurs co-transcriptionally in close temporal association with pre-mRNA splicing, or B) the microRNA hairpin is located in an exon of a long ncRNA (for example a long ncRNA transcribed from its own promoter). The former would necessarily occur in the nucleus, while the latter could conceivably occur in the cytoplasm.

Reviewer #3

(Remarks to the Author)

The manuscript here describes the function of Pasha’s WW domain in regulating Microprocessor localization and assembly in *C. elegans*. Through a forward genetic screen using a carefully designed and functional reporter of microprocessing activity, the authors uncovered a cis-regulatory mutation in the primary sequence of the sensor and a mutation near the WW domain of PASHA. They found that the WW domain of Pasha promotes its dimerization and nuclear retention and is critical for proper assembly of Microprocessor. This is an interesting, well-written work that established a fully functional sensor of Microprocessor activity and thoroughly characterizes the function of Pasha’s WW domain. We have only minor suggestions for revision.

- In figure 1C, the “10-fold lower” levels of the sensor could also be attributed to the differential promoter strengths between the endogenous miR-58 promoter and the ubl promoter used in the transgene. Please mention this alternative possibility.
- How many genomes were screened? Was the screen done to saturation?
- Using more standard nomenclature of the G and W mutant would be helpful, i.e. G179R. The GW-mut nomenclature is fine.
- On page 8, “The lack of correlation we observed between primary and mature miR-58 suggests that the amount of pri-miR-58 exceeds what can be processed by the miRNA machinery.” Another explanation for this discrepancy could be that a downstream step in biogenesis is rate limiting (such as dicing or loading); this would dampen the effect of changes in a non-rate-limiting step (microprocessing) on the mature microRNA levels.
- Please add a very short description in the methods of how iso-mirs are called by TinyRNA default mode.
- The authors demonstrate that pri-miRNA of miR-58 is stabilized in the G-mut (Figure 2E-F). After characterizing additional miRNAs affected by this mutation (Figure 3), no additional pri-miRNAs is are examined. Checking if pri-miRNAs of some of the other down-regulated miRNAs are stabilized would be informative.
- In figure 3A and 3B, it was not clear if there is a threshold for log2FC. The authors mentioned the p value and reads number cutoffs, but not the log2FC cutoff. Please clarify.
- Please add to the methods section what size interval was selected after PCR during small RNA library construction. If this interval excludes pre-miRNA reads, then the following statement should be removed or qualified with the caveat that these are contaminating reads outside the intended size selection interval. “Although there was a slight increase in pre-miRNA reads in G-mutants grown at 20°C, it was diminished at 25°C (Fig. 3D).”

- In figure 4E, 4F, 7B and 7F, it would be useful if the authors could quantify the nuclear and cytoplasmic fractions as they did in figure 4D. Especially considering that the different mutants (G-mut, W-mut, N-del and NES) exhibit varying degrees of phenotype severity.
 - In figure 6A-D, it would be helpful if the bands could be annotated to indicate expected bands and non-specific bands.
 - In figure 7, only PASH-1::GFP localization was characterized in the NES mutant. It would be interesting and informative to also examine the Drosha localization in the mutant.
- Figure 2F and H, y axis label missing 25C
 - Line 260, "implement" should probably be "implicate"
 - Line 491, "then" should be "than"

Version 1:

Reviewer comments:

Reviewer #1

(Remarks to the Author)

The key argument here is whether the mislocalization of Drosha in the Pasha dimerization mutant background is mainly explained by the transportation role of Pasha or by the Drosha protein's folding/solubility issue. Also, the more important one is whether this question is biologically important.

Hydrophobic interactions are the basic principle of protein folding. Hydrophobic residues should find their partner residues to fold a tertiary structure of a protein or make a protein complex (i.e., protein-protein interaction). Here, the authors found mutations that disrupt the homodimeric interface of Pasha. The exposed hydrophobic surface residues should find other hydrophobic residues to hide these residues from water. If the concentration of this protein is high, it will make soluble aggregates, as shown in the example of recombinant protein purification. If the concentration of this protein is not high, as in this example, it will misfold the protein. In this paper, the authors showed that the expression level of Pasha was reduced in the WW domain mutants, indicating that Pasha misfolding occurred. Then, how about the Drosha protein? Compared to other Western blots performed in the Pasha mutants, the Drosha protein level in the size-exclusion chromatography was significantly lower. Because the authors cannot see anything in the void fraction, it means that many misfolded Drosha proteins, which have exposed hydrophobic residues, were stuck to the resin during the chromatography. Therefore, I don't think the WW mutant data is useful in supporting their conclusion.

In contrast, I agree that the NES-fused Pasha data support their conclusion. The strong NES signal can trap the Microprocessor's nuclear import. However, I am not convinced why it is important to know this because a stable protein complex can easily be imagined moving together in cells.

In terms of the novelty, the authors provided 4 major points.

- (1) The first live, whole-animal sensor for pri-miRNA processing: Even the authors, the original inventors, could not find exciting things using this reporter system. I'm not sure whether other groups will enthusiastically welcome this system for their research.
- (2) A conserved GW motif within PASH-1's WW domain that is essential for PASH-1 dimerization and Microprocessor integrity: G179 is just the next residue to W180. G179's function should be the same as W180 according to the structure.
- (3) PASH-1 and DRSH-1 are mutually dependent for each other's subcellular localization: I discussed this above.
- (4) The first in vivo characterization of the C. elegans Microprocessor complex: The co-IP MS data just confirmed the expected composition of Microprocessor.

I do not support its publication in Nat Commun.

Reviewer #2

(Remarks to the Author)

In my judgement, the authors have satisfactorily addressed all the Reviewers' critiques, through additional experiments and revisions to the text. I am persuaded that the revised manuscript does contain enough novel data to be published in Nature Communications.

Reviewer #3

(Remarks to the Author)

Lines 345-347:

"Similarly, mCherry::DRSH-1 co-IP'd more efficiently with non-mutant PASH-1::GFP than with either the G179R or W180A mutant forms (Supplemental Fig. S6A-S6D)."

The authors should quantify these data; they are not convincing by eye because the recovery of the mutant versions of PASH-1::GFP is much lower than that of the wild type version. For this statement to be true, the ratio of IP'd mCherry::DRSH-1 to IP'd PASH-1::GFP should be lower in the PASH-1::GFP mutants than wild type by quantification. If not, please modify this sentence to reflect that the ratio of mCherry::DRSH-1 to PASH-1::GFP recovered is similar across genotypes, or that the outcome is inconclusive.

My previous comments have been adequately addressed.

We appreciate the thoughtful and constructive feedback from the reviewers. We have made substantial revisions to address the shortcomings identified in the original version. These are detailed in our point-by-point responses below. We also revised the text extensively for clarity and conciseness. We believe these revisions have significantly improved the manuscript and better highlight the impact and novelty of our findings.

Reviewers' comments:

Reviewer #1 (Remarks to the Author):

As protein aggregation is the major concern of reviewer 1, we will first address this issue before going through the comments point by point.

The reviewer argues that our findings regarding DRSH-1 nuclear localization in *pash-1* WW domain mutants may result from protein aggregation, drawing a parallel to observations of DROSHA fragments overexpressed in human cells without compensatory DGCR8 overexpression. The argument is based on an artificial overexpression system with inherent caveats that do not reflect the endogenous levels of full-length, genomically encoded proteins in our study. To counter this concern, we provide four lines of evidence:

1. Absence of Aggregates in Imaging Data

We do not observe mCherry::DRSH-1 puncta in any of the *pash-1* mutant lines, as might occur if aggregates were present. Instead, the fluorescence signal is very uniform, consistent with monomeric forms of the proteins or complexes (e.g. Fig. 5A):

Page 12, line 260 in the revised manuscript...

"It is possible that the G179R and W180A mutations cause PASH-1 to misfold or aggregate, which in turn could cause DRSH-1 to aggregate, as DROSHA aggregates when overexpressed in human cells in the absence of compensatory DGCR8 overexpression. We did not observe an increase in PASH-1::GFP or mCherry::DRSH-1 puncta in these mutants, suggesting that the proteins maintain their diffuse cellular expression and do not form large aggregates."

2. No Large Aggregates in Size Exclusion Chromatography

We do not observe large aggregates of DRSH-1 or PASH-1 by size exclusion chromatography (see Fig. 6A with void fractions now indicated):

Page 15, line 335 in the revised manuscript...

*"We did not observe large aggregates in the void fractions (Fig. 6A). However, we observed a shift in the elution profile of mCherry::DRSH-1 toward higher molecular weight fractions in *pash-1*[W180A]:GFP mutants (Fig. 6A). While this shift could suggest a tendency for small-scale oligomerization or conformational changes, the absence of a similar shift in *pash-1*[G179R]:GFP mutants indicates that it is unlikely to be the underlying cause of mislocalization in these mutants (Fig. 6A)."*

3. Nuclear Export Signal (NES) Experiments Support Complex Dependency

DRSH-1 always tracks with PASH-1 in the nucleus, regardless of whether we use RNAi knockdown, WW domain mutants, or N-terminal deletion mutants. However, each of these approaches may interfere with Microprocessor assembly, confounding the results. Therefore, we introduced *pash-1::GFP::NES* into animals containing *mCherry::drsh-1*. Because the nuclear export signal (NES) is a very small peptide (10 aa) fused to the C terminus of GFP, it is unlikely to interfere with protein folding or complex formation. In animals containing both *pash-1::GFP::NES* and *mCherry::drsh-1*, PASH-1::GFP::NES and mCherry::DRSH-1 are both excluded from the nucleus, suggesting that PASH-1::GFP::NES drives mCherry::DRSH-1 into the cytoplasm and demonstrating that DRSH-1 does not dissociate from PASH-1 to enter or remain in the nucleus (see Fig. 7H):

Page 18, line 415...

“The ectopic NES on PASH-1::GFP drove both PASH-1::GFP and mCherry::DRSH-1 to strongly mislocalize to the cytoplasm, demonstrating that DRSH-1 does not dissociate from PASH-1 and enter or remain in the nucleus outside of the Microprocessor complex (Fig. 7H).”

4. Ectopic Nuclear Localization Signals (NLS) Partially Rescue Nuclear Localization

In a major revision to the manuscript, we demonstrate that by introducing ectopic nuclear localization signals (NLSs) onto PASH-1 or DRSH-1 we could partially rescue their nuclear enrichment in the *pash-1* WW domain mutants. This indicates that proteins are not trapped in the cytoplasm solely because of aggregation:

Page 12, line 260 in the revised manuscript...

*“It is possible that the G179R and W180A mutations cause PASH-1 to misfold or aggregate, which in turn could cause DRSH-1 to aggregate, as DROSHA aggregates when overexpressed in human cells in the absence of compensatory DGCR8 overexpression^{14,40}. We did not observe an increase in PASH-1::GFP or mCherry::DRSH-1 puncta in these mutants, suggesting that the proteins maintain their diffuse cellular expression and do not form large aggregates. Given that PASH-1 and DRSH-1 are relatively large proteins (85 and 125 kD respectively), even small aggregates could impair nuclear import and contribute to the loss of nuclear enrichment in WW domain mutants. However, if we could artificially drive nuclear localization of PASH-1::GFP and mCherry::DRSH-1 in G179R and W180A mutants, this would suggest that aggregation alone does not underlie the loss of nuclear enrichment. To test this, we added sequences encoding EGL-13 and SV40 nuclear localization signals (NLSs) onto either end of GFP in *pash-1::GFP* and mCherry in *mCherry::drsh-1* in wild-type and WW domain mutant backgrounds, as these two NLSs cause strong nuclear enrichment of *C. elegans* proteins⁴¹. PASH-1::GFP and mCherry::DRSH-1 containing the two ectopic NLSs did not show enhanced nuclear enrichment in wild-type backgrounds (Supplemental Fig. S5B-S5C). Nevertheless, the PASH-1 WW domain mutants with the ectopic NLSs displayed a partial restoration of nuclear enrichment. (Supplemental Fig. S5B-S5C). Similarly, mCherry::DRSH-1 with the ectopic NLSs also displayed partial rescue of nuclear enrichment in the mutant backgrounds, however, with both proteins rescue was variable and never complete (Supplemental Fig. S5B-S5C). While these results do not rule out aggregation, they suggest that the cytoplasmic fractions of PASH-1 and DRSH-1 in the WW domain mutants retain the capacity to localize to the nucleus. This implies that protein aggregation is not solely responsible for the loss of nuclear enrichment in these mutants.”*

AND

Page 13, line 287...

“We next tested whether the two ectopic NLSs introduced into PASH-1::GFP and mCherry::DRSH-1 could rescue their nuclear localization following reciprocal RNAi knockdown of drsh-1 and pash-1. The ectopic NLSs on mCherry::DRSH-1 partially restored its nuclear enrichment following pash-1 RNAi, and the ectopic NLSs on PASH-1::GFP similarly led to partial rescue of its nuclear enrichment following drsh-1 RNAi (Supplemental Fig. S5F). These results indicate that DRSH-1 and PASH-1 can still localize to the nucleus in each other's absence when nuclear transport is artificially enhanced. That nuclear localization was only partially restored by the ectopic NLSs underscores the importance of proper Microprocessor formation in promoting nuclear entry or retention of PASH-1 and DRSH-1.”

In summary, our data collectively indicate that the nuclear mislocalization of DRSH-1 in pash-1 WW domain mutants arises from disrupted Microprocessor assembly rather than protein aggregation. This is supported by uniform fluorescence patterns, size exclusion chromatography, NES-driven cytoplasmic co-localization, and partial rescue by ectopic NLSs.

Montgomery et al. established a pri-miRNA processing reporter system in *C. elegans* and found that a G179R mutation in the WW domain of Pasha resulted in a pri-miRNA processing defect. The authors confirmed that the effect of the G179R mutation was similar to that of the W180A mutation, which is important in protein folding and dimerization of Pasha. Lastly, the authors claimed that the WW domain is required for the nuclear localization of the Drosha/Pasha complex.

Thank you for your insightful feedback, which has helped us to put forth a much improved manuscript.

Major points

Because the WW domain mutations affect the protein folding and dimerization of Pasha, it is not convincing that the WW domain plays a specific role in the subcellular localization of Drosha/Pasha. I interpret the results as the accumulation of soluble aggregated Drosha and Pasha proteins in the cytoplasm where translation occurs.

We apologize for the confusion. We do not believe that the WW domain is directly required for the nuclear localization of the complex. Instead, our findings demonstrate that the assembly of the complex is necessary for the nuclear localization of its individual components. We have revised the manuscript to make this point clearer. Specifically, we updated the title, abstract, and multiple sections within the results and discussion to better articulate our conclusions. For example...

TITLE

See page 1...

*“Interdependence of Pasha and Drosha for localization and function of the Microprocessor in *C. elegans*”*

ABSTRACT

Page 2, line 9...

“We find that the WW domain also facilitates nuclear localization of Pasha, likely through its role in Microprocessor assembly, which in turn promotes nuclear import or retention of Drosha.”

INTRODUCTION

Page 4, line 50...

“PASH-1 dimerization and Microprocessor integrity is impaired in WW domain mutants, which likely underlies the strong reduction in nuclear enrichment of both PASH-1 and DRSH-1 in these mutants. Knockdown of pash-1 via RNA interference (RNAi) also disrupts DRSH-1's nuclear enrichment, while drsh-1 knockdown leads to a reduction in PASH-1's nuclear enrichment. Thus, correct assembly of the Microprocessor is important for nuclear import or retention of both PASH-1 and DRSH-1.”

RESULTS

Page 10, line 222...

“However, the strong nuclear-to-cytoplasmic enrichment we observed for PASH-1::GFP was lost in both the G179R and W180A mutants, indicating that these residues are important, presumably indirectly, for nuclear entry or retention of PASH-1 (Fig. 4F-4G).”

DISCUSSION

Page 20, line 477...

“In the model that emerges from this study, Microprocessor integrity, facilitated in part by PASH-1 dimerization via the GW motif in the WW domain, is important for its nuclear entry or retention.”

It is known that the hydrophobic interaction between the second RNase III domain of human DROSHA and the C-terminal tail of DGCR8 is critical for DROSHA stability (Cell 164:81, 2016; ref. 42). Without this specific masking, human DROSHA protein becomes soluble aggregates (Cell 161:1374, 2015; ref. 14). I superimposed the AlphaFold2 predicted structure of *C. elegans* Drosha on the crystal structure of human DROSHA (PDB: 5B16). The second RNase III domain of *C. elegans* Drosha preserves surface hydrophobic residues (e.g., L920 and I967, corresponding to L1194 and V1243 of human DROSHA, respectively). Therefore, it seems that *C. elegans* Drosha would be aggregated (but still soluble) without the functional Pasha protein.

Aggregation of human DROSHA in the absence of masking by DGCR8 has been observed specifically in overexpression systems (Cell 161:1374, 2015; ref. 14 and Cell 164:81, 2016; ref. 42). It was noted (data not shown, Cell 161:1374, 2015; ref. 14) that human DROSHA aggregates heavily in HEK293E cells when overexpressed, which can be prevented by co-expression of full-length DGCR8. Conversely, DROSHA aggregates in *E. coli* and baculovirus systems were not prevented by DGCR8 co-expression (data not shown, Cell 161:1374, 2015; ref. 14), demonstrating that aggregation is context-dependent. Furthermore, to assess aggregation the studies expressed only a fragment of DROSHA (D3; residues 390–1365) with short fragments of DGCR8 (G1; residues 728-750 and G2; residues 701-773). In contrast, our study investigates DRSH-1 and PASH-1 produced from their endogenous loci in a whole organism context (*C. elegans*), allowing us to explore their interactions under native regulatory

and physiological conditions. We're not saying that these previous findings are incorrect, but you certainly can't conclude that aggregation will occur in a more native system based on what was observed in these unnatural contexts.

Then, I superimposed the AlphaFold2 predicted structure of *C. elegans* Pasha on the crystal structure of the human DGCR8 dimerization domain (PDB: 3LE4). G179R may affect the loop's flexibility and hinder the proper positioning of the side chain of F196. W180A will lose the critical hydrophobic dimerization interface. Therefore, the structural prediction implies that the WW domain mutations should affect Pasha's dimerization.

This was our conclusion as well, and we confirmed this by demonstrating that the PASH-1 dimerization region associates with full length unaltered PASH-1 but not with the G179R or W180A mutants (see Fig. 6D, unchanged from original version).

We added the AlphaFold3 structure prediction for the Microprocessor to demonstrate the similarity in *C. elegans* and humans (see Fig. 2E, 7B, and 7F).

The authors showed that the WW domain mutations affected the protein folding and dimerization of Pasha and the assembly of the Microprocessor (Fig. 6B-D). Thus, without the need for a special nuclear localization mechanism through the WW domain, we can interpret the data through protein misfolding and aggregation in the cytoplasm.

As noted above, we do not believe that the WW domain is directly responsible for nuclear localization of the Microprocessor. And as we describe in detail above, protein misfolding and aggregation cannot fully explain the loss of nuclear enrichment. One further line of evidence worth noting here is that Pasha appears to have endogenous roles that are independent of Drosha, suggesting that Pasha can still function in the absence of Drosha, and important considering our finding that DRSH-1 is also required for nuclear localization of PASH-1 (*PNAS* 111:1421–1426, 2014; ref. 54).

mCherry-Drosha is enriched near 669 kDa and 158 kDa in the presence of wild-type Pasha, and both peak positions were shifted toward the left (larger size) in the W-mut animal, indicating *C. elegans* Drosha became aggregated without a properly folded/dimerized Pasha (Fig. 6A).

While there does appear to be a very minor shift for W180A mutants, this is not observed in G179R mutants. Because both mutants show a similar DRSH-1 mislocalization phenotype, this cannot explain our results:

Page 16, line 361...

“However, we observed a shift in the elution profile of mCherry::DRSH-1 toward higher molecular weight fractions in pash-1[W180A]::GFP mutants (Fig. 6A). While this shift could suggest a tendency for small-scale oligomerization or conformational changes, the absence of a similar shift in pash-1[G179R]::GFP mutants indicates that it is unlikely to be the underlying cause of mislocalization in these mutants (Fig. 6A).”

(2) The manuscript does not contain enough novel data to be published in Nat Commun. As the authors mentioned, the pri-miRNA processing reporter is not conceptually new (PNAS 111:1861, 2014; ref. 23). The processing deficiency of the cis-acting mutation found in the stem region of a pri-miRNA can be expected (Mol Cell 60:131, 2015). The trans-acting factor screening found only one Pasha mutant, whose molecular consequence can be interpreted with protein misfolding and aggregation.

Firstly, we introduce the first live, whole-animal sensor for pri-miRNA processing, representing a significant advancement over prior studies in *Drosophila*. This novel approach enables imaging-based visual screening in live animals, leveraging the genetic tractability of *C. elegans* to provide a powerful and versatile platform for future research.

Secondly, we identify a conserved GW motif within PASH-1's WW domain that is essential for PASH-1 dimerization and Microprocessor integrity. This finding underscores the critical structural role of the WW domain in maintaining complex functionality.

Thirdly, we demonstrate that PASH-1 and DRSH-1 are mutually dependent for each other's subcellular localization. This key conceptual advance highlights that intact Microprocessor complex formation is essential for nuclear entry or retention of its components. While we did not identify a post-transcriptional regulatory mechanism between PASH-1 and DRSH-1 as seen in mammals, we reveal that crossregulation of their protein localization may serve a similar role in maintaining nuclear equilibrium.

Finally, this work provides the first in vivo characterization of the *C. elegans* Microprocessor complex, offering novel insights and valuable resources to drive further exploration of the miRNA pathway under physiologically relevant conditions.

Together, these findings deepen our understanding of Microprocessor assembly and function, as well as the intricate interplay between its components in regulating pri-miRNA processing.

Minor points

(1) Line 82. Because the pri-miRNA processing event occurs in the nucleus, I don't think the translational machinery affects this reduction.

There are various potential mechanisms through which competition between miRNA biogenesis and mRNA translation might occur in the nucleus. For instance, the exon junction complex could obstruct access to or processing by the Microprocessor. However, this is purely speculative, and we have revised this section as follows:

Page 5, line 75...

"The sensor restored miR-58 levels in mir-58^{-/-} mutants to ~11% of wild-type levels but had a negligible impact on miR-58 levels in mir-58^{+/+} animals, indicating that the miR-58 hairpin is further processed by the miRNA pathway following Microprocessor cleavage, albeit at much lower levels than the endogenous miR-58 gene (Fig. 1C). The sensor has the advantage over other sensors in that it reports on pri-miRNA processing in whole, live animals and can be scored on a standard stereo microscope."

(2) Line 91. I think an internal loop is a more appropriate term than a bulge.

Corrected.

(3) Line 148. The authors need to consider the kinetic difference between pri-miRNA and mature miRNA. A pri-miRNA is rapidly degraded because of DROSHA processing, but a mature miRNA is very stable.

We removed speculation over the reason behind the lack of correlation between pri-miRNA and mature miRNA levels.

(4) Line 179. Because the G and W residues seem to work on the same function(i.e., dimerization), the term 'independent' may confuse readers.

We revised the text as follows:

Page 8, line 155...

“This suggests that G179 and W180 contribute additively to the WW domain’s function.”

(5) Line 191. The authors can show the secondary structures of pri-miR-1829 and pri-miR-42 and explain why they are not optimal substrates for Microprocessor.

The miR-1829 gene family members have bulges at the sight of cleavage and unusually high levels of mispairing in the basal regions of the stems. We added the secondary structures to Figure 3, to illustrate this. See figure 3D and associated text:

Page 8, line 158...

“miR-1829 family members may be poor Microprocessor substrates due to bulges at their hairpin cleavage sites and large loops in their lower stems (Fig. 3D).”

(6) Fig 6A. The Pasha expression level of G- and W-mut are much lower than other Western blots. Are they all in void fractions?

We did not observe the G179R or W180A mutant proteins in the void fractions of our SEC. Nor were we able to detect them in the pelleted fractions of our lysates (not shown). In addition to the reduction we observed by Western blot, we see a consistent reduction in PASH-1::GFP fluorescence in animals containing these mutations. We believe that PASH-1 is less stable in these mutants possibly because they fail to dimerize correctly. We did not observe a reduction in PASH-1::GFP expression following drsh-1 knockdown, indicating that association with Drosha is not likely required for Pasha’s stability (see Fig. 5B-5D).

(7) Related to 6. Where are the void fractions? Based on Superdex 200's resolution, I think the first 3-5 fractions would be the void fractions.

We now indicate which lanes contain void fractions based on experimentally determined protein elution profiles (see Fig. 6A).

(8) Fig 4G. Protein levels of Drosha and Pasha need to be shown to exclude the possibility of other interpretations, such as transcriptional inhibition of the Drosha gene.

We can't examine mCherry::DRSH-1 and PASH-1::GFP levels in the cdk-9 mutant strain because it's balanced by GFP and we don't have endogenous antibodies. Also, the strain is early larval lethal making it difficult to work with. As we don't draw any major conclusions from this result, we don't believe additional experimentation is justified. These results are now supplemental. See Supplemental Figure S4 and associated text:

Page 11, line 232...

"We observed a modest increase in mCherry expression in first generation animals homozygous mutant for cdk-9, relative to their heterozygous counterparts, although loss of cdk-9 leads to larval arrest which could confound these results (Supplemental Fig. S4A). While this suggests that cdk-9 may have a role in pri-miRNA processing, because CDK-9 regulates Pol II activity, desilencing of the sensor could also be caused by reduced expression of pash-1 or drsh-1, which we did not explore further."

(9) Line 300. There are two hairpins in the DGCR8 mRNA. One is in the 5' UTR, and another is in the CDS.

Corrected.

Page 13, line 296...

"DGCR8 is downregulated through Microprocessor cleavage of two hairpins in its mRNA, while DROSHA is stabilized through protein-protein interactions involving DGCR8"

(10) Line 371. typo, 'in in'

Corrected.

(11) Line 450. The authors mentioned the possibility of other Drosha or other Pasha complexes. However, they had already done co-IP and MS experiments and could not find any other specific proteins.

AND...

(12) Line 460 and 490. W180 may not have an additional role except for Pasha dimerization according to its structure (PDB: 3LE4) and MS experiment.

In the original co-IP-MS experiments done by the Kim lab, they also did not identify additional protein interactions that came out later using more sensitive approaches. It is quite possible, if not likely that the Microprocessor interacts with other proteins, possibly in a more transient manner:

Page 16, line 372...

“This suggests that DRSH-1 and PASH-1 are the only major stable components of the Microprocessor in C. elegans, although more transient interactors may also exist.”

Page 20, line 457...

“The W180 residue may have additional roles, such as facilitating transient interactions important for miRNA processing. These roles may involve heme binding or protein orientation on miRNA hairpins, but further investigation is needed, as our mass spectrometry analysis did not detect such interactions.”

Page 20, line 478...

“It will be important in future studies to determine if the WW domain is important for interactions with other proteins in C. elegans using more sensitive techniques than those applied here.”

Reviewer #2 (Remarks to the Author):

Montgomery et al present findings related to the function of the WW domain of the Pasha (DGCR8) component of the microprocessor, which also includes the RNaseIII enzyme Drosha. The WW domain of Pasha had been previously shown to be necessary for Pasha dimerization, which is necessary for assembly of functional microprocessor — which consist of a Pasha dimer associated with a Drosha monomer. The authors show that mutations at two conserved amino acids of the C. elegans Pasha WW domain compromise microprocessor activity in vivo resulting in reduction of many microRNAs and consequent developmental phenotypes. Biochemical analysis of in vivo complexes, and fluorescence microscopy of tagged Drosha and Pasha in vivo, show that the WW domain mutations affect the nuclear localization of both the mutant Pasha protein and also Drosha.

Thus one of the central findings of the manuscript is a role for Pasha in facilitating Drosha's correct localization. Interestingly, the levels of expression of Pasha and Drosha reciprocally affect each others' nuclear localization. The authors show that depletion of Pasha by RNAi caused mCherry::DRSH-1 to accumulate primarily in the cytoplasm. Similarly, RNAi-knockdown of Drosha caused PASH-1::GFP to localize cytoplasmically. The authors tested whether the reciprocal mutual regulation of Pasha and Drosha reflects an indirect consequence of compromised microRNA activity, and found that RNAi knockdown of alg-1 and alg-2 did not affect Pasha and Drosha localization to the nucleus, consistent with a relatively direct mutual regulation.

A noteworthy contribution of the manuscript is the demonstration of a microprocessor sensor consisting of a transgene expressing a fluorescent protein (FP) from an mRNA containing a microRNA hairpin in its 3' UTR; normal microprocessor activity efficiently excises the hairpin from the reporter mRNA, destabilizing the mRNA and preventing FP expression; expression of the FP reveals compromised microprocessor activity on the reporter, enabling forward screens for mutations that compromise microprocessor activity. This reporter was used to isolate the WW mutation that serves as the basis for this study, which augurs well for the value of the reporter for future genetic studies of microprocessing in C. elegans.

Overall, the manuscript present important findings about the interdependence of microprocessor components in *C. elegans* for the proper assembly, cellular localization, and function of microprocessor. A specific significant contribution of the manuscript is the demonstration of the importance of certain amino acids in the Pasha WW domain for microprocessor assembly. A broader significant contribution is the implication of subunit stoichiometry in the dynamics of microprocessor cytoplasmic trafficking.

Thank you for your valuable insight!

Minor critiques:

Line 81: "This reduction may be attributed to the translational machinery sequestering the transcript away from the miRNA pathway due to the presence of the mCherry coding sequence." This is confusing because the translational machinery would be in the cytoplasm, so it could not interfere with processing by microprocessor, if microprocessor is nuclear. Explain.

Reviewer 1 also raised this concern. There are various potential mechanisms through which competition between miRNA biogenesis and mRNA translation might occur in the nucleus. For instance, the exon junction complex could obstruct access to or processing by the Microprocessor. However, this is purely speculative, and in streamlining text in the revised manuscript we excluded this point.

Line 93: "Heterozygous cis-acting mutations that disrupt cleavage of one copy of the sensor should partially de-silence mCherry expression, whereas only dominant or dosage-dependent heterozygous mutations in trans-acting factors would lead to de-silencing." This sentence is confusing as worded. It would be helpful to clarify that the context is the F1 self-progeny population from mutagenized hermaphrodites.

AND...

Line 107: "We draw two conclusions from these results. Firstly, the recognition or processing of the miR-58 hairpin is susceptible to slight perturbations in the miRNA duplex region. Secondly, the sensor accurately reflects the recognition and processing of the miR-58 hairpin by the Microprocessor machinery." The reporter itself is a valuable tool, but that is revealed by the recovery of the Pasha mutant. The cis-acting mutation in the reporter is not very informative, and raises more questions than it answers, such as why was it that only one out of 500 worms with de-silenced transgene contained a mutation. Also, the modest de-repression observed upon reconstructing that mutation is concerning. It would be better to remove from the narrative the presentation of the cis-screen.

We moved the cis-acting mutant from the main figure to Supplemental Fig. 1. We revised the text to streamline the results with a focus on their value in validating that the sensor is responsive to mutations in the hairpin, which we believe is important enough to warrant including these results.

See revised Supplemental Figures S1A-S1B and revised text:

Page 5, line 81...

“In a proof-of-principle forward genetic screen targeted at identifying cis-acting mutations that disrupt sensor processing, we identified a mutation within the miRNA duplex region of the hairpin (Supplemental Fig. S1A). The mutation led to a modest increase in mCherry expression, which we confirmed by introducing it into non-mutagenized animals using site-directed mutagenesis (Supplemental Fig. S1B). The mutation changes a G-C base pair to a G-U base pair adjacent to the internal loop in the hairpin, likely causing the internal loop to expand and thereby preventing recognition or processing by the Microprocessor. That this modest perturbation to the hairpin prevents its processing, demonstrates that the sensor accurately reflects the recognition and cleavage of the miR-58 hairpin by the Microprocessor machinery.”

147: “The lack of correlation we observed between primary and mature miR-58 suggests that the amount of pri-miR-58 exceeds what can be processed by the miRNA machinery.” The wording of this sentence is somewhat confusing. Perhaps the authors mean that microprocessing is not the rate-limiting biogenesis step in the accumulation of mature mir-58.

Reviewer 1 had a similar concern. We removed speculation over the reason behind the lack of correlation between pri-miRNA and mature miRNA levels:

Page 7, line 117...

“Similar results were obtained without normalization (Fig. 2I). Unlike pri-miR-58, mature miR-58 levels were similar between 20°C and 25°C in wild-type animals, suggesting a non-linear relationship between primary and mature miR-58 levels (Fig. 2I).”

Line 176: “Surprisingly animals containing both mutations produced significantly fewer progeny than either of the single mutants” More explanation is required about why this is surprising.

Good point. We’ll leave it to the readers to decide whether this is surprising. It was unexpected to us because we predicted that the W residue mutation would completely abolish the function of the domain and thus additional mutations to the domain would be inconsequential. We revised the text as follows:

Page 7, line 143...

“Although the W residue mutated here is a defining feature of WW domains, we investigated whether the G179 and W180 residues, collectively referred to as the GW motif, have additive functions by generating a strain with both mutations (GW-mut). These double mutants produced significantly fewer progeny than either single mutant (Supplemental Fig. S2E; mutations were made in the pash-1::GFP background described below). This suggests that G179 and W180 contribute additively to WW the domain’s function.”

Line 185: “At 25°C, a less ideal growth condition for C. elegans.... “ This is a rather vague statement; less ideal in what regard?

To simplify and streamline the text, we removed the text asserting that 25°C is a less ideal growth condition.

Line 192: “We did not confirm that the miRNAs identified in Ahmed et al. are authentic miRNAs, but it is possible that these are also non-Microprocessor dependent miRNAs as they were also upregulated.” Confusing. More explanation seems to be required. What is this statement in reference to? Mir-42 is a microRNA.

We revised this section as follows:

Page 8, line 155...

“Mirtrons, which bypass Microprocessor processing, were upregulated in mutants at both temperatures (Fig. 3C; Supplemental Data 1)29,30. Additionally, the miR-1829 family (4 miRNAs) and miR-42 cluster (2 miRNAs) were upregulated at 20°C (Fig. 3C; Supplemental Data 1). However, the miR-42 cluster was depleted at 25°C, while the miR-1829 family was elevated at both temperatures (Supplemental Data 1). miR-1829 family members may be poor Microprocessor substrates due to bulges at their hairpin cleavage sites and large loops in their lower stems (Fig. 3D). If these miRNAs are Microprocessor-independent, their processing mechanism is unclear. Several miRNAs identified by Ahmed et al. were also upregulated and thus may also bypass Microprocessor processing (Fig. 3C; Supplemental Data 1).”

Line 195: “...mutants there is reduced competition for shared resources downstream of pri-miRNA processing due to the likely reduction in pre-miRNAs, which are subsequently diverted to mirtrons and other Microprocessor-independent miRNAs.” Addition explanation seems to be required to clarify that “upregulated” miRs could be those that are relatively less dependent than others on wt pasha.

We revised this section as follows:

Page 8, line 162...

“Misprocessing of pri-miRNAs in pash-1(ram33[G179R]) mutants could free downstream resources, such as Dicer or Argonaute, which may be redirected to mirtrons and other Microprocessor-independent miRNAs. Conversely, miRNAs that are less sensitive to the G179R mutation but still dependent on the Microprocessor could be upregulated due to reduced competition with other pri-miRNAs.”

Line 260: “...miR-58 sensor, which would implement it in pri-miRNA processing.” ‘implement’ should be “implicate”.

Corrected.

Line 455: “Interestingly, the W mutant is sick, whereas the G mutant is relatively healthy.” Please define ‘sick’ vs ‘healthy’. Are these qualitative difference or quantitative?

See the following revision:

Page 19, line 453...

“Despite a crucial role for PASH-1 in C. elegans development, mutations within its WW domain are relatively well tolerated. While the W180A mutation caused severe developmental defects and reduced fertility, the neighboring G179R mutation caused only mild defects and near-normal fertility, even though both residues are similarly required for DRSH-1 interaction and nuclear localization.”

Line 471: “Therefore, it is possible that, unlike in mammals, C. elegans miRNAs are predominantly processed post-transcriptionally.” The definition ‘post-transcriptionally’ in this context is a bit ambiguous. Perhaps the salient distinction would be whether A) the microRNA hairpin is located in an intron of an mRNA such that microprocessing occurs co-transcriptionally in close temporal association with pre-mRNA splicing, or B) the microRNA hairpin is located in an exon of a long ncRNA (for example a long ncRNA transcribed from its own promoter). The former would necessarily occur in the nucleus, while the latter could conceivably occur in the cytoplasm.

We revised this text as follows, although I’m not sure we fully addressed your concern:

Page 20, line 463...

“It is possible that, unlike in humans, C. elegans miRNAs are processed after transcription and splicing are completed. While most miRNAs occur in the introns of protein-coding or non-coding genes in humans, only a small subset are produced from presumptive introns in C. elegans. Even the introns that contain miRNAs in C. elegans appear to have regulatory elements distinct from the host transcript and thus may not undergo excision, which could be important for pri-miRNA processing if it occurs after transcription, since introns are typically degraded during splicing. Additionally, introducing an intron into a miRNA hairpin does not disrupt processing of the hairpin or alter the mature miRNA sequence in C. elegans, indicating that splicing occurs before pri-miRNA processing. In contrast, in humans, pri-miRNAs spanning intron-exon junctions are largely excluded from processing by the Microprocessor. Regardless of timing, our results demonstrate that excluding the Microprocessor from the nucleus leads to embryonic or early larval lethality, emphasizing a critical requirement for nuclear processing. While some pri-miRNA processing in the cytoplasm is possible, it likely cannot compensate for loss of nuclear processing, perhaps because pri-miRNAs are not efficiently exported to the cytoplasm or because handoff to the Dicer complex is restricted to the nucleus”

Reviewer #3 (Remarks to the Author):

The manuscript here describes the function of Pasha’s WW domain in regulating Microprocessor localization and assembly in C. elegans. Through a forward genetic screen using a carefully designed and functional reporter of microprocessing activity, the authors uncovered a cis-regulatory mutation in the primary sequence of the sensor and a mutation near the WW domain of PASHA. They found that the WW domain of Pasha promotes its dimerization and nuclear retention and is critical for proper assembly of Microprocessor. This is an interesting, well-written work that established a fully functional sensor of Microprocessor activity and thoroughly characterizes the function of Pasha’s WW domain. We have only minor suggestions for revision.

We appreciate your valuable suggestions!

- In figure 1C, the “10-fold lower” levels of the sensor could also be attributed to the differential promoter strengths between the endogenous miR-58 promoter and the ubl promoter used in the transgene. Please mention this alternative possibility.

Good point. Here’s the revised text after some streamlining to reduce the word count:

Page 5, line 75...

“The sensor restored miR-58 levels in mir-58^{-/-} mutants to ~11% of wild-type levels, but had a negligible impact on miR-58 levels in mir-58^{+/+} animals, indicating that the miR-58 hairpin is further processed by the miRNA pathway following Microprocessor cleavage, albeit at much lower levels than the endogenous miR-58 gene (Fig. 1C).”

- How many genomes were screened? Was the screen done to saturation?

The screen was not done to saturation and only one line was characterized. We included additional details in the RESULTS section as follows:

Page 5, line 91...

“We then did a forward genetic screen for trans-acting mutations that desilenced the pri-miR-58 sensor. From a non-exhaustive screen of ~40,000 haploid genomes, we selected 72 candidates representing at least 20 independent lines that desilenced mCherry. Of these, only 17 were fertile over multiple generations. Among the fertile lines, 38a displayed the most robust increase in mCherry expression (Fig. 2A-2B). We subjected this line to whole genome sequencing and identified a missense mutation in pash-1 (Fig. 2C).”

We also describe the mutagenesis in the METHODS section:

Page 22, line 520...

“Forward genetics

L4 and young adult C. elegans were rotated in M9 buffer solution containing 1% ethyl methyl sulfonate (EMS) solution for 4 hours at room temperature and then washed 4x in M9. For each experiment, 50 mutagenized P0 animals were plated onto 20 10 cm NGM plates seeded with E. coli OP50. In the F1 screen for cis-acting factors (which identified the pri-miR-58 mutation), 500 bright fluorescent animals were selected from among ~100,000 F1 animals. In the F2 screen (which identified the pash-1(ram33[G179R]) mutation) up to 4 bright fluorescent individuals were selected from each of 20 distinct pools of F2 animals, each derived from ~2,000 F1s. For the F1 screen, individual F1 animals were lysed and the miR-58 hairpin sequence of the pri-miR-58 sensor was sequenced using Sanger sequencing. For the F2 screen, individual animals were isolated and then expanded for several generations. The brightest healthy line after several generations was subjected to whole genome sequencing to identify the causal mutation.”

- Using more standard nomenclature of the G and W mutant would be helpful, i.e. G179R. The GW-mut nomenclature is fine.

We revised the manuscript and figures as suggested and following WormBase guidelines.

- On page 8, “The lack of correlation we observed between primary and mature miR-58 suggests that the amount of pri-miR-58 exceeds what can be processed by the miRNA machinery.” Another explanation for this discrepancy could be that a downstream step in biogenesis is rate limiting (such as dicing or loading); this would dampen the effect of changes in a non-rate-limiting step (microprocessing) on the mature microRNA levels.

We reworded this section to remove speculation as to why there is a lack of correlation:

Page 6, line 117...

“Unlike pri-miR-58, mature miR-58 levels were similar between 20°C and 25°C in wild-type animals, suggesting a non-linear relationship between primary and mature miR-58 levels (Fig. 2I).”

- Please add a very short description in the methods of how iso-mirs are called by TinyRNA default mode.

Please see the revised text and the newly added Supplemental Data 4 which contains the tiny-count selection rule parameters:

Page 9, line 172 of the RESULTS section...

“However, we did not observe an increase in the abundance of miRNA isoforms (iso-miRs) shifted by 1-3 nucleotides (nts) at their 5’ ends relative to the annotated miRNA locus in pash-1(ram33[G179R]) mutants (Fig. 3A-3B; Supplemental Data 1).”

AND IN THE METHODS...

Page 25, line 587...

“To identify iso-miRs, selection rules were defined within tiny-count to capture reads with 5’ ends aligning 1-3 nucleotides upstream or downstream of the 5’ end of the annotated miRNA. ... (Supplemental Data 4).”

- The authors demonstrate that pri-miRNA of miR-58 is stabilized in the G-mut (Figure 2E-F). After characterizing additional miRNAs affected by this mutation (Figure 3), no additional pri-miRNAs are examined. Checking if pri-miRNAs of some of the other down-regulated miRNAs are stabilized would be informative.

We assayed 4 additional pri-miRNAs at both 20° and 25°C. Each of them was also upregulated. See supplemental figure S2A-S2B and revised text:

Page 6, line 119...

“pri-miR-35, pri-miR-51, pri-miR-80, and pri-miR-238 were also all significantly upregulated in mutants at both 20°C and 25°C (Supplemental Fig. S2A-S2B).”

- In figure 3A and 3B, it was not clear if there is a threshold for log₂FC. The authors mentioned the p value and reads number cutoffs, but not the log₂FC cutoff. Please clarify.

We did not apply a fold-change cutoff. While we appreciate the importance of such cutoffs, particularly for assessing biological relevance, they are of course inherently arbitrary. In this case, since we are not focusing on conclusions regarding individual miRNAs, applying a fold-change cutoff would have limited value. For example, applying a 1.5-fold cutoff to the bar plot in Figure 3A would only exclude 19 downregulated miRNAs. However, omitting these miRNAs from the analysis would not alter our overall conclusion. In the revised manuscript, we note the fold-change range observed:

Page 8, line 161...

“At 20°C, most canonical miRNAs were significantly downregulated (1.2-16.5-fold) in mutants relative to wild-type (Fig. 3A; Supplemental Data 1).”

- Please add to the methods section what size interval was selected after PCR during small RNA library construction. If this interval excludes pre-miRNA reads, then the following statement should be removed or qualified with the caveat that these are contaminating reads outside the intended size selection interval. “Although there was a slight increase in pre-miRNA reads in G-mutants grown at 20°C, it was diminished at 25°C (Fig. 3D).”

We added the size range as noted below. But more importantly, we edited the results section to clarify that we examined small RNA reads derived from pre-miRNAs, not the pre-miRNAs themselves:

METHODS

Page 24, page line 583...

~134-148 nt small RNA PCR amplicons, corresponding to ~16-30 nt small RNAs, were size selected on a 10% polyacrylamide gel, transferred by electrophoresis to DE81 chromatography paper, eluted at 70°C for 20 minutes in the presence of 1 M NaCl, and precipitated at -80°C overnight in the presence of 13 ug/ml glycogen and 67% EtOH.”

RESULTS

Page 9, line 176...

“We detected only minor differences in sRNA-seq reads originating from pre-miRNA sequences that neither correspond to mature miRNAs nor to the iso-miRs described above (Fig. 3A-3B; Supplemental Data 1).”

- In figure 4E, 4F, 7B and 7F, it would be useful if the authors could quantify the nuclear and cytoplasmic fractions as they did in figure 4D. Especially considering that the different mutants (G-mut, W-mut, N-del and NES) exhibit varying degrees of phenotype severity.

Unfortunately, the high background green fluorescent signal we observe in wild type animals lacking GFP would cause this to be a highly inaccurate and potentially misleading calculation. This is a common problem with GFP in *C. elegans*, especially when working with endogenously

tagged loci with low-to-moderate expression. Note that in in Fig. 4E we only quantified mCherry::DRSH-1, which is not confounded by this issue because there is essentially no background red fluorescence and it is consistent between embryos.

- In figure 6A-D, it would be helpful if the bands could be annotated to indicate expected bands and non-specific bands.

We added asterisks next to the expected bands. See revised Fig. 6A-6D.

- In figure 7, only PASH-1::GFP localization was characterized in the NES mutant. It would be interesting and informative to also examine the Drosha localization in the mutant.

Absolutely! We made the strain and observed that mCherry::DRSH-1 is also strikingly mislocalized to the cytoplasm. See Fig. 7H and associated text:

Page 18, line 415...

“The ectopic NES on PASH-1::GFP drove both PASH-1::GFP and mCherry::DRSH-1 to strongly mislocalize to the cytoplasm, demonstrating that DRSH-1 does not dissociate from PASH-1 and enter or remain in the nucleus outside of the Microprocessor complex (Fig. 7H).”

- Figure 2F and H, y axis label missing 25C

Thanks for catching that.

- Line 260, “implement” should probably be “implicate”

We corrected the typo. Thanks.

- Line 491, “then” should be “than”

Thanks!

REVIEWER COMMENTS

Reviewer #1 (Remarks to the Author):

The key argument here is whether the mislocalization of Drosha in the Pasha dimerization mutant background is mainly explained by the transportation role of Pasha or by the Drosha protein's folding/solubility issue. Also, the more important one is whether this question is biologically important.

Hydrophobic interactions are the basic principle of protein folding. Hydrophobic residues should find their partner residues to fold a tertiary structure of a protein or make a protein complex (i.e., protein-protein interaction). Here, the authors found mutations that disrupt the homodimeric interface of Pasha. The exposed hydrophobic surface residues should find other hydrophobic residues to hide these residues from water. If the concentration of this protein is high, it will make soluble aggregates, as shown in the example of recombinant protein purification. If the concentration of this protein is not high, as in this example, it will misfold the protein. In this paper, the authors showed that the expression level of Pasha was reduced in the WW domain mutants, indicating that Pasha misfolding occurred. Then, how about the Drosha protein? Compared to other Western blots performed in the Pasha mutants, the Drosha protein level in the size-exclusion chromatography was significantly lower. Because the authors cannot see anything in the void fraction, it means that many misfolded Drosha proteins, which have exposed hydrophobic residues, were stuck to the resin during the chromatography. Therefore, I don't think the WW mutant data is useful in supporting their conclusion.

In contrast, I agree that the NES-fused Pasha data support their conclusion. The strong NES signal can trap the Microprocessor's nuclear import. However, I am not convinced why it is important to know this because a stable protein complex can easily be imagined moving together in cells.

In terms of the novelty, the authors provided 4 major points.

- (1) The first live, whole-animal sensor for pri-miRNA processing: Even the authors, the original inventors, could not find exciting things using this reporter system. I'm not sure whether other groups will enthusiastically welcome this system for their research.
- (2) A conserved GW motif within PASH-1's WW domain that is essential for PASH-1 dimerization and Microprocessor integrity: G179 is just the next residue to W180. G179's function should be the same as W180 according to the structure.
- (3) PASH-1 and DRSH-1 are mutually dependent for each other's subcellular localization: I discussed this above.
- (4) The first in vivo characterization of the *C. elegans* Microprocessor complex: The co-IP MS data just confirmed the expected composition of Microprocessor.

I do not support its publication in Nat Commun.

Response:

We appreciate the reviewer's thoughtful and detailed critique of our manuscript. Below, we address each of the main concerns raised and clarify the significance of our findings, although it is unlikely that this reviewer will be persuaded.

Mislocalization of Drosha in the Pasha WW domain mutant background

The reviewer raises an important point about whether the observed mislocalization of Drosha is primarily due to Pasha's role in transport or a Drosha folding/solubility issue. While we cannot entirely rule out protein misfolding, the nuclear mislocalization of Drosha in Pasha mutants occurs independently of major solubility issues. Our SEC and Western blot data show that while some protein loss occurs, Drosha is still detectable in relevant fractions, suggesting that nuclear localization is not solely dictated by solubility constraints.

We explicitly note the possibility of protein misfolding in the revised manuscript...

Page 12, line 299:

“It remains possible that PASH-1 misfolds in these mutants, potentially causing misfolding of DRSH-1 as well, however, we did not investigate this possibility further.”

And...

Page 12, line 307:

“Therefore, PASH-1 and DRSH-1 are required for each other’s nuclear localization, but whether the mechanism relates to a cooperative role in nuclear entry or retention or to a role in proper protein folding is unclear.”

We further note here that because the complex still forms to some extent in PASH-1 WW domain mutants and retains the ability to process pri-miRNA, albeit less efficiently, this indicates that the PASH-1 mutant proteins are at least partially still folded properly.

Significance of the NES-fused Pasha experiment

We acknowledge the reviewer’s perspective that stable protein complexes often move together within cells. However, our NES-Pasha data provide direct experimental evidence that nuclear import or retention of Drosha is influenced by Pasha’s localization, rather than just an assumed property of stable complex formation. This helps distinguish whether Pasha acts passively as a binding partner or plays a more active role in controlling localization of Drosha, a finding that remains relevant to understanding Microprocessor function and potentially independent roles for Drosha and Pasha.

Novelty and impact of our findings

(1) Whole-animal sensor for pri-miRNA processing: our reporter system enables spatiotemporal analysis of pri-miRNA processing at single-cell resolution in a living organism. This represents an advance over in vitro assays and may prove useful for future studies on tissue-specific regulation of pri-miRNA processing.

(2) Conserved GW motif and its function: We agree that because G179 is adjacent to W180 it likely contributes similarly to dimerization, but our findings demonstrate that disrupting these specific residues impacts Pasha dimerization and function, providing new insights into Microprocessor assembly. Furthermore, because mutations in each of these residues lead to different developmental phenotypes but have similar subcellular localization, it remains possible that they do indeed have distinct functionality.

(3) Mutual dependency of PASH-1 and DRSH-1 localization: As discussed earlier, we provide several lines of evidence suggesting that their localization is not solely dictated by folding-related stability. Furthermore, if the proteins are incapable of folding correctly or if they aggregate in the absence of one another, it is unclear how the many Microprocessor independent roles that have been reported for the proteins are accomplished.

(4) *In vivo* characterization of the *C. elegans* Microprocessor complex: While our co-IP MS data confirm the expected composition, they provide a whole organism analysis of Microprocessor interactions, distinct from previous studies. This contextual information is valuable for understanding how the complex might function in different tissues and further highlights that Drosha and Pasha are the major stable components of the complex in whole organisms.

While we appreciate the reviewer’s constructive feedback, we believe our findings contribute meaningful insights into Microprocessor biology and provide new tools for the field.

Reviewer #2 (Remarks to the Author):

In my judgement, the authors have satisfactorily addressed all the Reviewers' critiques, through additional experiments and revisions to the text. I am persuaded that the revised manuscript does contain enough novel data to be published in Nature Communications.

Reviewer #3 (Remarks to the Author):

Lines 345-347:

“Similarly, mCherry::DRSH-1 co-IP'd more efficiently with non-mutant PASH-1::GFP than with either the G179R or W180A mutant forms (Supplemental Fig. S6A-S6D).”

The authors should quantify these data; they are not convincing by eye because the recovery of the mutant versions of PASH-1::GFP is much lower than that of the wild type version. For this statement to be true, the ratio of IP'd mCherry::DRSH-1 to IP'd PASH-1::GFP should be lower in the PASH-1::GFP mutants than wild type by quantification. If not, please modify this sentence to reflect that the ratio of mCherry::DRSH-1 to PASH-1::GFP recovered is similar across genotypes, or that the outcome is inconclusive.

Response:

We quantified the blots as suggested. See page 14, line 366 of the revised manuscript:

“Wild-type PASH-1::GFP co-IP'd with mCherry::DRSH-1 effectively, but this interaction was significantly diminished in PASH-1 G179R mutants (Fig. 6b and Supplementary Fig. S6a, b). Similarly, mCherry::DRSH-1 co-IP'd more efficiently with non-mutant PASH-1::GFP than with the G179R mutant form (Supplementary Fig. S6a, b). Therefore, the G179 residue contributes to the assembly or stability of the Microprocessor. We also observed a trend of reduced interaction between PASH-1::GFP and mCherry::DRSH-1 in the W180A form of PASH-1. However, due to high variability among biological replicates, possibly resulting from developmental defects in pash-1[W180A] mutants, this difference was not statistically significant (Fig. 6c and Supplementary Fig. S6c, d).”

My previous comments have been adequately addressed.